# Hearing of malaria mosquitoes is modulated by a beta-adrenergic-like octopamine receptor which serves as insecticide target

Marcos Georgiades [1,2,10], Alexandros Alampounti [1,2,10], Jason Somers[1,2], Matthew P. Su [1,2,3,4], David A. Ellis [1,2], Judit Bagi[1,2], Daniela Terrazas-Duque[1], Scott Tytheridge[1], Watson Ntabaliba[5], Sarah Moore [5,6,7,8], Joerg T. Albert [1,2,9] ✉ & Marta Andrés [1,2] ✉

Malaria mosquitoes acoustically detect their mating partners within large swarms that form transiently at dusk. Indeed, male malaria mosquitoes preferably respond to female flight tones during swarm time. This phenomenon implies a sophisticated context- and time-dependent modulation of mosquito audition, the mechanisms of which are largely unknown. Using transcriptomics, we identify a complex network of candidate neuromodulators regulating mosquito hearing in the species *Anopheles gambiae*. Among them, octopamine stands out as an auditory modulator during swarm time. In-depth analysis of octopamine auditory function shows that it affects the mosquito ear on multiple levels: it modulates the tuning and stiffness of the flagellar sound receiver and controls the erection of antennal fibrillae. We show that two α- and β-adrenergic-like octopamine receptors drive octopamine's auditory roles and demonstrate that the octopaminergic auditory control system can be targeted by insecticides. Our findings highlight octopamine as key for mosquito hearing and mating partner detection and as a potential novel target for mosquito control.

Sensory organs extract information from the environment. As the environment is dynamic, sensory organs have evolved mechanisms that enable sensory plasticity to adapt their physiology to environmental changes. The acoustic detection of mating partners in swarms by malaria mosquitoes constitutes a superb example of the adaptation of a sensory organ -the mosquito ear- to a transient change in the sensory ecology. Malaria mosquito swarms are brief and transitory aggregations of up to a thousand mosquitoes that take place every

sunset[1–3]. Within the swarm, mosquitoes are exposed to an acoustically challenging, noisy environment. It is against this noisy acoustic backdrop that male mosquitoes identify and locate the flight tones of their female mating partners[4–6].

Across insects, a remarkable and unique feature of the Johnston's organ (JO), the mosquitoes' 'inner ear', is its efferent innervation[7]. Input from efferent neurons from the brain seems to be an intrinsic component of mosquito audition, similar across evolutionarily distant

[1]Ear Institute, University College London, 332 Gray's Inn Road, London WC1X 8EE, UK. [2]The Francis Crick Institute, 1 Midland Road, London NW1 1AT, UK. [3]Graduate School of Science, Nagoya University, Nagoya, Aichi 464-8602, Japan. [4]Institute for Advanced Research, Nagoya University, Nagoya, Aichi 464-8601, Japan. [5]Vector Control Product Testing Unit (VCPTU), Environmental Health and Ecological Sciences, Ifakara Health Institute, P.O. Box 74 Bagamoyo, Tanzania. [6]Swiss Tropical and Public Health Institute, Socinstrasse 57, PO Box, CH-4002 Basel, Switzerland. [7]University of Basel, Petersplatz 1, CH-4001 Basel, Switzerland. [8]The Nelson Mandela African Institution of Science and Technology (NM-AIST), P.O. Box 447 Tengeru, Arusha, Tanzania. [9]Cluster of Excellence Hearing4all, Sensory Physiology & Behaviour Group, Department for Neuroscience, School of Medicine and Health Sciences, Carl Von Ossietzky University Oldenburg, Carl Von Ossietzky Str. 9-11, 26111 Oldenburg, Germany. [10]These authors contributed equally: Marcos Georgiades, Alexandros Alampounti. ✉e-mail: joerg.albert@uni-oldenburg.de; marta.andres@ucl.ac.uk

mosquito species[8]. The efferent fibres release neurotransmitters, including the biogenic amines octopamine, and serotonin, and the inhibitory neurotransmitter GABA[7]. Efferent activity may hold the key to the extraordinary performance of mosquito ears. Indeed, ablating efferent signalling causes the onset of male self-sustained oscillations (SSOs)[8], believed to act as amplifiers of female wingbeats in the swarm and to be essential for its detection[8]. We hypothesize that efferent-related (and other) neuromodulators tailor the performance of malaria mosquito audition to mating partner detection in the swarm.

A putative efferent neuromodulator that serves an auditory function is octopamine. *An. gambiae* males present phonotactic responses towards female flight tones while swarming, when they also erect their antennal fibrillae[9] – a phenomenon believed to increase the male sensitivity to female flight tones. In *An. stephensi* mosquitoes, fibrillae erection is induced by octopamine[10]. In other insects, octopamine modulates a plethora of behaviours and senses[11,12]: it conveys circadian clock information to sensory organs, modulating pheromone mating responses in moths[13–16] and attraction to conspecific secreted volatiles during the locust gregarious phase[17]. Moreover, as octopamine signalling is mostly restricted to invertebrates[11,18], it is a valid target for insecticide development[19–21]. Indeed, octopamine receptors are the only G-protein coupled receptors (GPCRs) targeted by insecticides, although they have not been exploited for mosquito control.

In this study, we profile the malaria mosquito ear transcriptome across the day. We identify a peak of expression of the α-adrenergic-like octopamine receptor *AgOAMB* (AGAP000045) at swarm time, and high levels of expression across the day of the β-adrenergic-like octopamine receptor, *AgOctβ2* (AGAP002886). We thoroughly investigate octopamine's role in *An. gambiae* audition and find that it mostly modulates male hearing, with lesser effects in females. Our results suggest that in males, octopamine has multimodal auditory effects: it controls the erection of the antennal fibrillae, sets flagellar stiffness, and modulates auditory tuning, presumably enhancing the male's ability to detect the female in the swarm. Moreover, we determine that *AgOctβ2* is the primary octopamine receptor in the mosquito ear as mutant males completely fail to erect their fibrillae and present minimal auditory changes upon octopamine injection. We also show that the octopaminergic signalling is activated by insecticides and is a valid target for mosquito population control. Together, our results suggest that octopamine acts as an important modulator of hearing in malaria mosquitoes, facilitating the detection of mating partners by tuning their auditory physiology to the swarm acoustic environment.

## Results

### Transcriptomics suggest a complex neuromodulatory control of mosquito audition

Mosquito ears are amongst the most complex sensory organs in insects; they are composed of two main elements, a sound receiver, the antennal flagellum, and an auditory sensory organ, the Johnston's organ (JO)[1]. To characterize the neuromodulatory network of *An. gambiae* ear and identify potential regulators during swarm time, we undertook RNA-sequencing analyses of male and female ears at six different circadian zeitgeber time (ZT) points throughout the day (including ZT12 or laboratory swarm time). We collected exclusively second antennal segments, hosting the JO, without flagella (Fig.1a). First, we conducted a computational procedure of read simulation, followed by Kallisto read quantification (Supplementary Data 1), that enabled us to estimate the noise distribution of read counts (Fig. 1b, see Methods). This noise distribution was employed in identifying transcripts that were present in the male, and female JOs (i.e. that had read counts above our estimated noise floor, with adjusted *p*-value ≤ 0.05). These are hereby referred to as *expressed* transcripts. Transcripts were independently identified in males and females (Supplementary Data 2 and 3, respectively) and assigned to GO accession numbers whose molecular function was potentially related to neuromodulation (Table 1; including biogenic

amines, classical neurotransmitters and neuropeptide receptors, as well as other GPCRs or receptors related to other sensory modalities). Within these categories, we identified 173 and 152 'neuromodulation genes' expressed in male and female JOs, respectively (Fig. 1c, d; Table 2; Supplementary Data 4 and 5). On a separate, and complementary analysis, we also looked for differential expression of transcripts between the male and female JO tissues (Table 2, Supplementary Data 6).

Biogenic amines are important sensory organ modulators in insects[16]. Our results also support this for the malaria mosquito ear as several biogenic amine receptors were expressed in both sexes (Fig. 1c, d). An ortholog of the octopamine β2 receptor in *Drosophila*, AGAP002886, was the highest expressed biogenic amine receptor, and had significantly higher expression in male than female JOs (Table 2). Other α and β octopamine receptors, serotonin receptors (including 5-HT1a, 5-HT1b, 5-HT2a, 5-HT2b and 5-HT7), a histamine receptor and a dopamine / ecdysone receptor were expressed. Many of them showed sexually dimorphic expression. We also examined the expression of classical neurotransmitter receptors. GABA acts as an efferent neurotransmitter in the ear of *Cx. quinquefasciatus* mosquitoes[7]. Multiple GABA receptors were expressed in both sexes including three metabotropic GABA-B receptor orthologs and the ionotropic GABA-A receptor *Rdl*, which causes resistance to the insecticide dieldrin in *Anopheles* populations[22]. We also found several nicotinic acetylcholine receptor subunits with sexually dimorphic expression. Nicotinic acetylcholine receptors are critical components of the efferent auditory system in vertebrates[23]. Moreover, a single muscarinic acetylcholine receptor was found, as were several metabotropic and ionotropic glutamate receptors, including an ortholog of *Drosophila* IR93a glutamate receptor (AGAP000256), involved in thermo- and hygrosensation[24].

Neuropeptides act as important neuromodulators of insect sensory neurons[25]. Our dataset included a broad repertoire of neuropeptide receptors (Table 2). Orthologs of the natriuretic peptide, the sex peptide, calcitonin and allatostatin 3 receptors were expressed in both sexes. We also identified putative tachykinin receptors 1 and 2, which have been implicated in the modulation of olfactory neurons in *Drosophila*[26], as well as several neuropeptide F receptor orthologs. Neuropeptide F modulates the locomotor plasticity of swarming migratory locusts[27]. Receptors involved in other sensory modalities were also identified, suggesting modulation of auditory responses following unconventional signalling pathways[28]. Of these, visual rhodopsins were particularly highly expressed. Rhodopsins have been previously shown to mediate auditory and mechanosensory roles in *Drosophila*[29,30]. Some gustatory receptors were expressed mostly in males, and two olfactory receptors were expressed in females.

We were particularly interested in the auditory modulation at swarm time. Swarm formation in mosquitoes is a rhythmic behaviour under circadian regulation. In the final step of our RNA-seq analysis, we identified transcripts with a rhythmic (or *cycling*) expression in the 24-hour period of data collection. Our full analysis is provided in Supplementary Data 7 and 8. From the dataset of expressed genes with a potential neuromodulatory role, there was one single transcript in males exhibiting cycling expression (Fig.1e): the octopamine α-receptor *AgOAMB* (AGAP000045), an ortholog of the OAMB receptor in *Drosophila*[31]; its expression peaked during the laboratory swarm time.

### Octopamine modulates malaria mosquito audition in a time-dependent, diel manner

Our transcriptomic analysis suggested the role of octopamine as an auditory modulator at swarm time. We explored its auditory role by injecting octopamine into the mosquito thorax and analysing changes in auditory function including (Fig. 2). In a nutshell, individual tests quantified different functionally relevant mechanical properties of the mosquito ear. It should be noted that all measures represent compound responses, i.e. all mechanical parameters (e.g. flagellar best frequency or tuning sharpness) reflect the sum total of all components

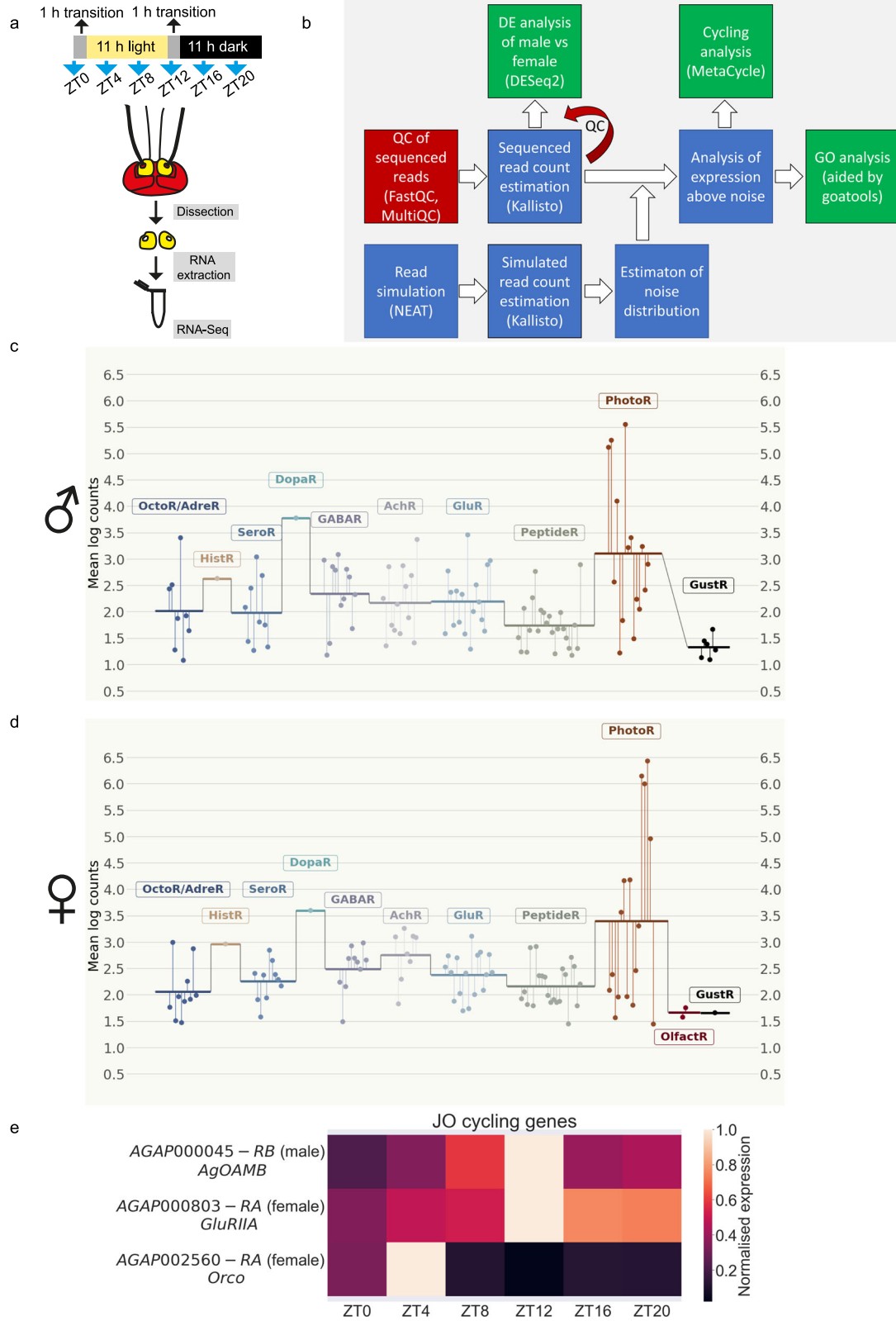

mechanically connected to the flagellum. Taken together these tests will enable better reconstructions, or predictions, of mosquito auditory sensitivity in different behavioural or ecological contexts. At present, it is e.g. not known in which contexts male flagellar ears enter the SSO state, respectively. Likewise, the sensory-ecological control of fibrillae erection is still not fully clear. Yet the state of all these parameters will have crucial effects on the sensitivity and tuning of the

mosquito ear. We thus assessed the pattern of fibrillae erection and quantified the properties of both stimulated and unstimulated ears. We used frequency sweeps to determine the auditory properties in response to more naturalistic stimuli and we used force step actuation as more analytic stimuli to extract principal parameters of flagellar mechanics (e.g. flagellar stiffness). We used two different octopamine concentrations (1 mM and 10 mM) to explore sensitivity and dynamic

**Fig. 1 | Transcriptomics of potential neuromodulatory genes in the JOs of male and female *An. gambiae* mosquitoes. a** Sample collection and preparation for RNA-sequencing. Mosquitoes were entrained to 12 h: 12 h light/dark cycle with one hour light transition between light and dark phases. Male and female mosquito were collected every four hours (lights on or ZT0, ZT4, ZT8, swarm time or ZT12, ZT16 and ZT20), their ears dissected and RNA extracted for RNA-sequencing. **b** Schematic workflow of the analysis of RNA-sequencing data (see Methods for more details). **c, d** Plots summarising the expression levels of transcripts belonging to selected GO categories in males (**c**) and females (**d**) potentially involved in neuromodulation. The transcriptomic data support the existence of an extensive neuromodulatory network acting on the mosquito JO. Each point represents a transcript that depicts the log of the mean expression counts of that transcript across all circadian time points. Horizontal lines represent the mean of the log

counts represented by each point within a GO category. **e** Heatmap representing normalised expression per time point of genes showing cycling expression in males and females from the "neuromodulatory" dataset. A single gene in males, the α-adrenergic-like octopamine receptor *AgOAMB*, and two genes in females, the odorant receptor co-receptor *Orco* (AGAP002560) and ionotropic glutamate receptor subunit GluRIIa (AGAP000803) show cyclic expression in the mosquito ear. JO: Johnston's organ. Abbreviations used in c and d: XXXR denotes the group of transcripts that show XXX receptor or XXX receptor-related activity. Octo = octopamine, Adre = adrenergic, Hist = histamine, Sero = serotonin, Dopa = dopamine, GABA = γ-Aminobutyric acid, Ach = acetylcholine, Glu = glutamate, Peptide = any peptide neurotransmitter, Photo = photoreceptors, Olfact = olfactory, Gust = gustatory.

range differences. We repeated the tests at two different circadian times: during a phase of inactivity for *An. gambiae* mosquitoes (ZT4) and during the laboratory swarm time (ZT12). Control injections were performed using a ringer solution.

**Octopamine caused erection of male antennal fibrillae.** A well-characterised circadian clock signature of *An. gambiae* audition is the rhythmic pattern of antennal fibrillae erection in males at swarm time[32,33], which is induced by octopamine in *An. stephensi* mosquitoes[10]. We tested whether this also applies to *An. gambiae* (Fig. 3).

A visual assessment confirmed that 100% of males presented erected fibrillae at ZT12 and 0% at ZT4 (Fig. 3a). All mosquitoes collapsed the fibrillae after mounting them in preparation for auditory tests, irrespective of the circadian time. This allowed us to study the effects of octopamine from a baseline state in which all mosquitoes (both at ZT4 and ZT12) had collapsed fibrillae. Injecting 1 mM octopamine caused the full erection of the antennal fibrillae in 33% of mosquitoes at ZT4 and 62.5% at ZT12 suggesting circadian time-dependent changes in the JO sensitivity to octopamine effects ($\chi^2$, p-value < 0.001). Increasing the octopamine concentration to 10 mM caused increased rates of fibrillae erection (ZT4: 60%, ZT12: 83%).

**Octopamine prompted a quiescent state in the male mosquito flagellum.** We then analysed the flagellar behaviour in males under unstimulated conditions. Male mosquito flagella have been reported to exhibit two different mechanical states. Some present spontaneous 'self-sustained oscillations' (SSOs) that are large oscillations of around 350 Hz (~ female wing-beat frequencies) and velocity magnitudes

around 1 mm/s, which is ~1000-fold above baseline levels[8]. By contrast, quiescent receivers show best frequencies of ~500 Hz and substantially lower velocity magnitudes (~1 μm/s). Although the ecological relevance of these two states is not fully understood, SSOs seem to act as amplifiers of the female wingbeats[5,6,8].

We first developed an analytical framework to provide a quantitative definition of the different mechanical states (Fig. 2d, see Methods). We noted that apart from being in a quiescent or SSO state, the mosquito flagellum could also display a transient state between the two (Fig. 3b). Control mosquitoes tend to maintain the same mechanical state for the duration of the experiment (Supplementary Fig. 1). We examined whether octopamine could induce changes in the mechanical state. Injecting 1 mM octopamine at ZT4 pushed transient mosquitoes into a quiescent state, while the proportion of SSO mosquitoes remained unaltered (Fig. 3b and Supplementary Fig. 1). The effect was stronger upon 10 mM injections. By contrast, at ZT12, injecting octopamine induced a shift from SSO to a quiescent state, as we observe a decrease in the number of SSO and a concurrent increase of quiescent mosquitoes. We conclude that octopamine drives the flagellum towards the quiescent state, and that this effect is circadian-time dependent. While at ZT4 octopamine affects only transient mosquitoes, at ZT12 the sensitivity to octopamine is higher and hence also SSO mosquitoes are pushed into quiescence. We cross-validate these findings by looking at the fraction of time spent in each mechanical state and find statistical significance across all groups of control, OA 1 mM and OA 10 mM between ZT4 and ZT12 ($\chi^2$, p-value < 0.001). Our data support that the effect of octopamine on the flagellar mechanical state changes over the day.

**Octopamine caused an increase in SSO frequency.** Free fluctuation recordings were used to extract the oscillator frequency, $f_O$, and oscillation amplitudes of the flagellum's spontaneous vibrations. Quiescent mosquitoes do not produce sinusoidal or periodic signals and a frequency could not be conventionally determined within the resolutions of our paradigms. For this reason, they are not represented here (see Methods). The oscillator frequency tended to be constant during the recording for single mosquitoes under control conditions (Supplementary Fig. 1).

Octopamine injections caused an overall increase in the SSO frequency and a decrease in the amplitude (Fig. 3c, Supplementary Tables 1 and 2, SSO frequency [ZT4]; control: 351 ± 7 Hz showing median ± median absolute deviation (mad); OA1mM: 518 ± 92 Hz, p.adj <0.001; SSO frequency [ZT12]; control: 356 ± 10 Hz; OA1mM: 398 ± 53 Hz, p.adj <0.001). This effect was more pronounced at ZT12 for 10 mM octopamine injections where the average SSO frequency values were above 500 Hz, compared to around 350 Hz for controls (p.adj <0.001). The damping effect of octopamine on the SSO amplitude was also stronger at ZT12 upon 10 mM octopamine injections (Fig. 3c, SSO amplitude [ZT4]; OA10mM: 420 ± 115 nm; SSO amplitude [ZT12]; OA10mM: 280 ± 68 nm, p.adj <0.001). The stronger responses to octopamine at ZT12 indicated an increase in both the sensitivity to

**Table 1 | GO term identities and definitions for the GO terms used in transcript classification**

| GO term | Definition |
| --- | --- |
| GO:0004989 | octopamine receptor activity |
| GO:0004952 | dopamine neurotransmitter receptor activity |
| GO:0004969 | histamine receptor activity |
| GO:0008226 | tyramine receptor activity |
| GO:0004930 | G protein-coupled receptor activity |
| GO:0008227 | G protein-coupled amine receptor activity |
| GO:0001653 | peptide receptor activity |
| GO:0015464 | acetylcholine receptor activity |
| GO:0016917 | GABA receptor activity |
| GO:0008066 | glutamate receptor activity |
| GO:0008066 | glutamate receptor activity |
| GO:0004984 | olfactory receptor activity |
| GO:0099589 | serotonin receptor activity |
| GO:0008527 | gustatory receptor activity |
| GO:0009881 | photoreceptor activity |

**Table 2 | Genes expressed in the male and female JO potentially related to auditory neuromodulation, including receptors for octopamine, histamine, dopamine, serotonin, GABA, acetylcholine, glutamate and peptides**

| | *An. gambiae* gene | Annotation / Closest annotated ortholog | Male read counts | Female read counts | | *An. gambiae* gene | Annotation / Closest annotated ortholog | Male read counts | Female read counts |
|---|---|---|---|---|---|---|---|---|---|
| OA receptor | **AGAP002886** | **Octβ2R** | **2543** | **989*** | Peptide receptors | AGAP003283 | NPRA | 786 | 516* |
| | **AGAP000045** | **Oamb** | **272** | **181*** | | AGAP010486 | AstA R | 827 | 392* |
| | AGAP000606 | Octα2R | 326 | 751* | | AGAP029618 | SPR | 249 | 826* |
| | AGAP002888 | Octβ3R | 140 | 125 | | AGAP004122 | NPF R3 | 180 | 792* |
| | | | | | | AGAP008702 | - | 185 | 228 |
| | AGAP002566 | Histamine R | 430 | 925* | | AGAP003654 | CL R3 | 96 | 347* |
| | | | | | | AGAP005229 | MIP R | 117 | 149 |
| | AGAP005681 | DopEcR | 6029 | 3990* | | AGAP009770 | CL R1 | 69 | 322* |
| | | | | | | AGAP003244 | CAPA R | 97 | 160* |
| Serotonin receptors | AGAP004222 | 5-HT7 R | 1102 | 703* | | AGAP002824 | TkR86C | 78 | 49* |
| | AGAP004223 | 5-HT7 R | 489 | 449 | | AGAP002156 | GnRH R1 | 47 | 759* |
| | AGAP011481 | 5-HT1B R | 282 | 148* | | AGAP001592 | TkR99D | 83 | 162* |
| | AGAP002229 | 5-HT2B R | 122 | 256* | | AGAP012378 | NPF R3 | 62 | 217* |
| | AGAP002232 | 5-HT2A R | 27 | 88* | | AGAP001962 | CAPA R1 | 47 | 243* |
| | AGAP007136 | 5-HT1A R | 19 | 38* | | AGAP004555 | - | 32 | - |
| | | | | | | AGAP000351 | NPF R1 | 56 | 81* |
| GABA receptors | AGAP010281 | GABA-B-R1 | 957 | 967 | | AGAP012268 | SST R | 49 | 198* |
| | AGAP004595 | GABA-B-R2 | 617 | 458* | | AGAP011452 | CCHa R1 | 43 | 65* |
| | AGAP006028 | Rdl (GABA-A R) | 459 | 490 | | AGAP010851 | Lkr | 41 | 114* |
| | AGAP000038 | GABA-A R | 133 | 497* | | AGAP001961 | CAPA R2 | 32 | 312* |
| | AGAP009514 | GABA-B-R3 | 48 | 174* | | AGAP003631 | CCHa R2 | 16 | 50* |
| Acetylcholine receptors | AGAP002152 | nAChRα6 | 2360 | 1295* | | AGAP004035 | LHCGR | 17 | 77* |
| | AGAP008588 | nAChRα5 | 718 | 1824* | | AGAP001558 | GnRH R2 | 17 | 72* |
| | AGAP000966 | nAChRβ1 | 750 | 1256* | | AGAP001773 | AstA R | 15 | 75* |
| | AGAP009493 | nAChRα9 | 489 | 1224* | Photoreceptors | AGAP013149 | ninaE / Rh2/ Rh6 | 358032 | 2724940* |
| | AGAP000329 | nAChRα3 | 304 | 198* | | AGAP012982 | ninaE / Rh2/ Rh6 | 179762 | 1409728* |
| | AGAP002972 | nAChRα2 | 180 | 599* | | AGAP012985 | ninaE / Rh2/ Rh6 | 132010 | 1003613* |
| | AGAP002971 | nAChRα | 140 | 428* | | AGAP010089 | Rh5 | 12599 | 91526* |
| | AGAP000138 | nAChRα4 | 103 | - | | AGAP001161 | ninaE / Rh2/ Rh6 | 1737 | 15177* |
| | AGAP002974 | nAChRα1 | 45 | - | | AGAP006126 | Rh3 | 2025 | 18316* |
| | AGAP000962 | nAChRα7 | 39 | 68* | | AGAP001162 | ninaE / Rh2/ Rh6 | 112 | 243* |
| | AGAP010513 | mAChR-A | 23 | 81 | | AGAP007548 | Rh7 | 99 | 93* |
| Glutamate receptors | AGAP005034 | mGluR | 940 | 242* | OR | **AGAP002560** | **Orco** | **-** | **57** |
| | AGAP008644 | mGluR | 398 | 1396* | | AGAP011991 | Or61 | - | 38 |
| | AGAP000256 | Ir93a | 596 | 504* | | | | | |
| | AGAP006027 | GluRIA | 215 | 645* | | | | | |
| | AGAP001434 | GluClα | 596 | 106* | | | | | |
| | AGAP002891 | mGluR | 324 | 266* | | | | | |
| | **AGAP000803** | **GluRIIA** | **236** | **338*** | | | | | |
| | AGAP012578 | mGluR | 103 | 104 | | | | | |
| | AGAP000801 | GLURIIB | 154 | 550* | | | | | |
| | AGAP029236 | mGluR | 58 | 146* | | | | | |
| Gustatory receptors | AGAP006713 | Gr | 47 | - | | | | | |
| | AGAP007757 | Gr | - | 46 | | | | | |
| | AGAP029169 | Gr | 28 | - | | | | | |
| | AGAP001137 | Gr59 | 24 | - | | | | | |
| | AGAP006717 | Gr26 | 19 | - | | | | | |
| | AGAP009805 | Gr9 | 12 | - | | | | | |

Receptors involved in other sensory modalities (olfaction, gustation and photoreception) are also shown. Only genes with higher read counts in each category are shown (complete dataset in Supplementary Data 3 and 4). Genes discussed in the manuscript are highlighted in bold. Read count corresponds to the normalized average across all time points of sample collection. * in the last column indicates that the gene was differentially expressed between males and females, $p < 0.05$. Gene names are italicized. *OA* octopamine, *OR* olfactory receptors.

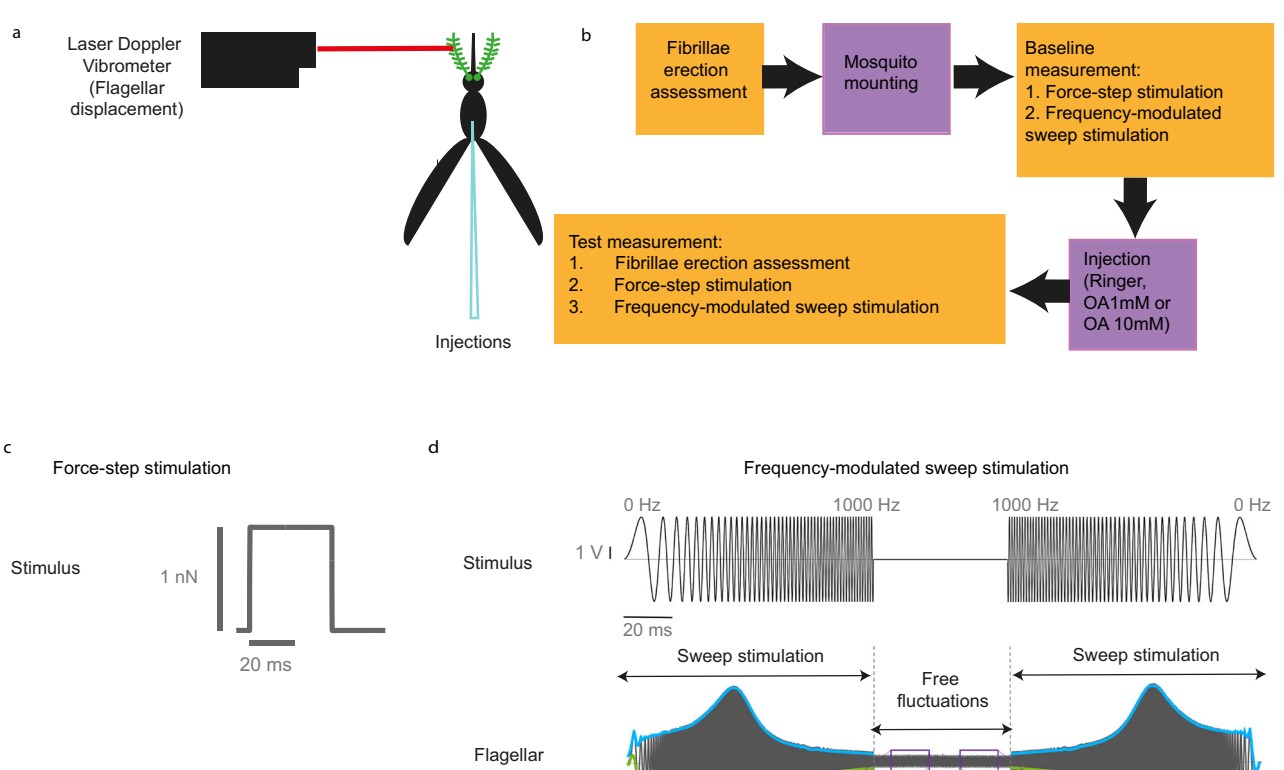

**Fig. 2 | Experimental setup and analysis pipeline for auditory analysis.**
**a** Experimental paradigm of Laser Doppler vibrometry (LDV) recordings. The laser beam from a Laser Doppler Vibrometer is pointed at the tip of the mosquito flagellum to record flagellar displacement. **b** Auditory tests working pipeline. The experiment started by assessing the fibrillae erection state of the mosquito. The mounting procedure consisted in gluing the mosquito to a rod, so that only the right flagellum was free to move. Baseline measurements were collected including responses to force-step and frequency-modulated sweep stimulation. After control or test injections, the fibrillae erection was assessed again, and flagellar responses to force-step and frequency-modulated sweep stimulation were recorded.
**c** Example of force-step stimulation. Force-step actuation was used to extract the flagellar steady-state stiffness (see Methods section for more information).

**d** Example of frequency-modulated sweep stimulation that was applied to the mosquito flagellum. The stimulus (top) consisted of a linear frequency sweep that ranges from 0 Hz to 1 kHz in 1 s and its mirrored version of 1 kHz to 0 Hz. Between both sweeps, flagella remained 1 s unstimulated (shown as a shorter section in the figure). The flagellar responses to sweeps (middle) were used to extract an "envelope" of the waveform, which was fitted to a driven harmonic oscillator function to extract different biophysical parameters (e.g. oscillator and peak frequencies, Q-factor, acceleration). The unstimulated sections (free fluctuations) were used to (i) to calculate the frequency and amplitude of the unstimulated flagellar vibrations via FFT, (ii) analyse the amplitude distribution to determine the flagellar mechanical state (bottom; see Methods section for more information).

octopamine and the dynamic range of octopamine effects. At ZT4, 10 mM injections did cause weaker responses than 1 mM (SSO frequency [ZT4]; OA1mM: 518 ± 92; OA10mM: 398 ± 53, Hz $p$.adj <0.001), suggesting that responses were already saturated.

**Octopamine affected the auditory tuning of stimulated male mosquito ears.** We stimulated the flagellum using frequency-modulated sweeps (upchirps and downchirps, 0–1000 Hz Fig. 2d) and measured the flagellar displacements (Fig. 4, Supplementary tables 3 and 4). We fitted forced damped harmonic oscillator models to the resulting flagellar responses and extracted the relevant biophysical parameters. We studied receivers in both quiescent and SSO mechanical states.

Octopamine injection affected the oscillator at multiple levels (Fig. 4a–d). The most striking effect was an increase in the peak and

oscillator frequencies both in quiescent and SSO animals that was circadian-time dependent, and overall, stronger at ZT12 upon 10 mM octopamine injections, indicating a higher sensitivity and an increased dynamic range during the laboratory swarm time (Fig. 4a,b, oscillator frequency SSO [ZT4]; control: 343 ± 42 Hz; OA10mM: 448 ± 112 Hz, $p$.adj <0.001; oscillator frequency SSO [ZT12]; control: 350 ± 17 Hz; OA10mM: 503 ± 16 Hz, $p$.adj <0.001).

**Octopamine caused an increase in the flagellar stiffness of male mosquitoes.** Changes in flagellar steady-state stiffness upon octopamine injections were calculated from force-step actuation[8] as an indication of the ear's baseline properties and their relative contribution to flagellar sensitivity (Fig. 2c). The flagellar steady-state stiffness describes the force required to hold the flagellum at a certain steady-state displacement. Injecting 1 mM octopamine caused a

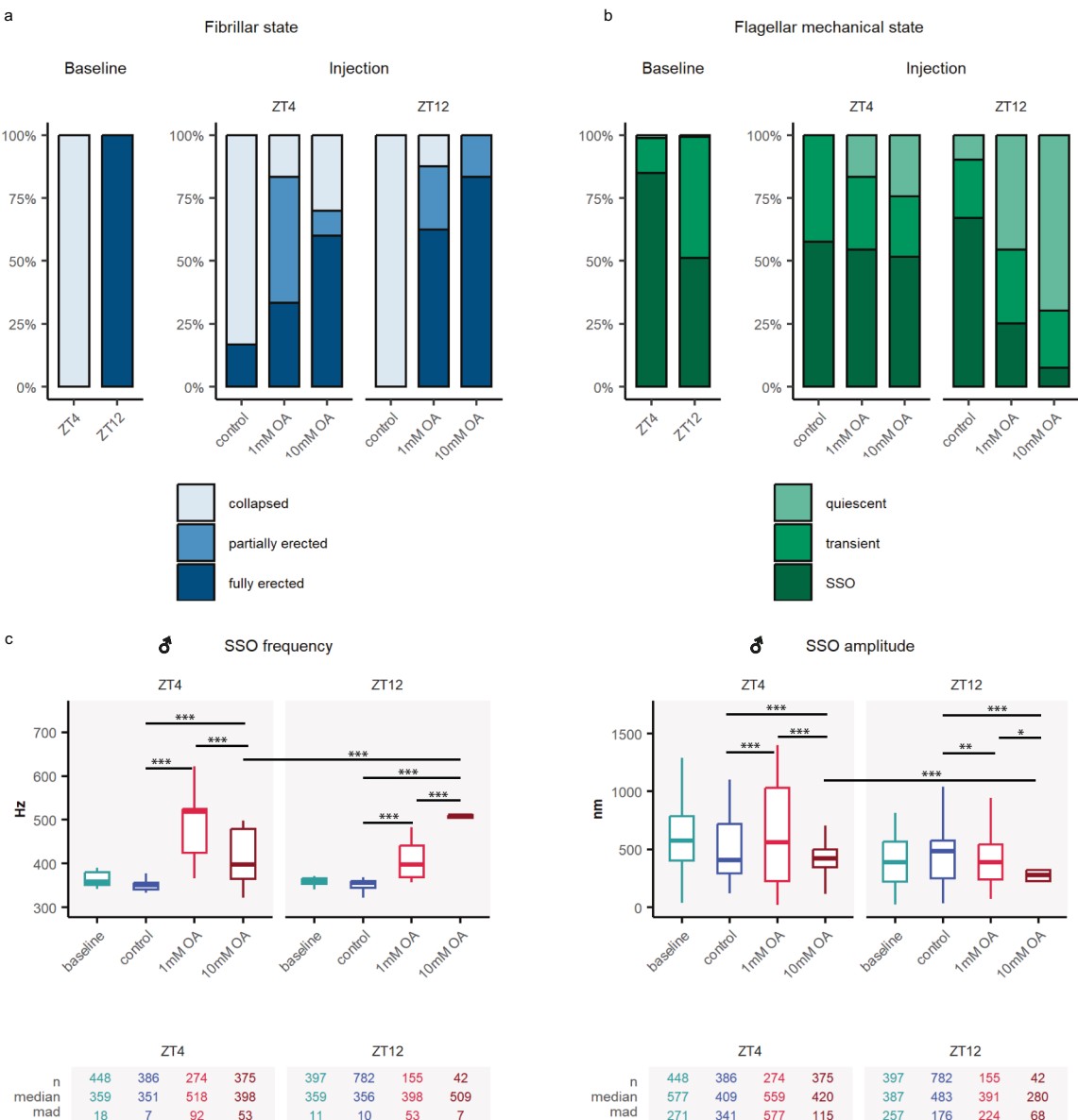

**Fig. 3 | Octopamine injection causes the erection of the antennal fibrillae and influences the flagellar mechanical state in male *An. gambiae* mosquitoes.** **a** Fibrillae erection state under different experimental conditions. Left panel, fibrillae erection baseline levels before mosquito mounting. At ZT12, during the laboratory swarm time, all male mosquitoes presented erected fibrillae, while at ZT4 -when *An. gambiae* mosquitoes are inactive-, all mosquitoes presented collapsed fibrillae. Right panel, fibrillae erection state upon octopamine injections. Please note that because of the mounting process, all mosquitoes collapsed their fibrillae, so before injecting octopamine all mosquitoes had collapsed fibrillae. Injecting both 1 mM and 10 mM octopamine caused the erection of the antennal fibrillae. The effect was stronger at ZT12 for 10 mM octopamine injections ($\chi^2$, *p*-value < 0.001). **b** Flagellar mechanical state based on the analysis of unstimulated free fluctuations of the flagellum. Left panel shows baseline states. Octopamine injection caused a shift to quiescent states and a sharp reduction in the

amount of SSO mosquitoes that was more pronounced at ZT12. **c** SSO frequency and amplitude in different experimental conditions. Octopamine injections caused a sharp shift in the SSO frequency to higher values and a decrease in the amplitude of the oscillations. Central line, median; box limits, first and third quartiles; lower and upper whiskers, 5th and 95th percentiles, respectively. Tables underneath each graph display n numbers (total number of runs (unstimulated sections) that passed the curation process and are included in the analysis), median and median absolute deviation (mad) for each category in the graph above. Significant differences between injection effects starred (two-sided Wilcoxon rank-sum tests with Holms procedure for multiple comparison correction, **p*.adj <0.05; ***p*.adj <0.01; ****p*.adj <0.001). OA octopamine, SSO self-sustained oscillations, ZT zeitgeber time. Sample sizes: ZT4 control = 8; ZT4 OA1mM = 8; ZT4 OA10mM = 7; ZT12 control = 11; ZT12 OA1mM = 9; ZT12 OA10mM = 11.

sharp increase in flagellar stiffness (Fig. 4e, Supplementary tables 5 and 6) that was circadian time-dependent and strongest at ZT12 upon 10 mM octopamine injections, where stiffness values were 7-fold higher than the control ([ZT12]; control: 195 ± 66 μN/m; OA10mM: 1258 ± 683 μN/m, *p*.adj <0.05).

**Octopamine injection had milder effects on female *Anopheles gambiae* mosquito audition.** Our transcriptomic data showed

expression of octopamine receptors also in female JO, although expression levels were lower and did not cycle along the day (Table 1, Fig. 1e). We examined the responses to frequency-modulated sweep and force-step stimulation, as described for males. Overall, injecting octopamine caused substantially smaller changes in the biophysical parameter values compared to males, although some changes were observed (Fig. 5a-e, e.g. peak frequency [ZT4] control: 420 ± 45.5 Hz; [ZT4] OA1mM: 383 ± 35.9 Hz, *p* < 0.001; [ZT12] control: 421 ± 36 Hz;

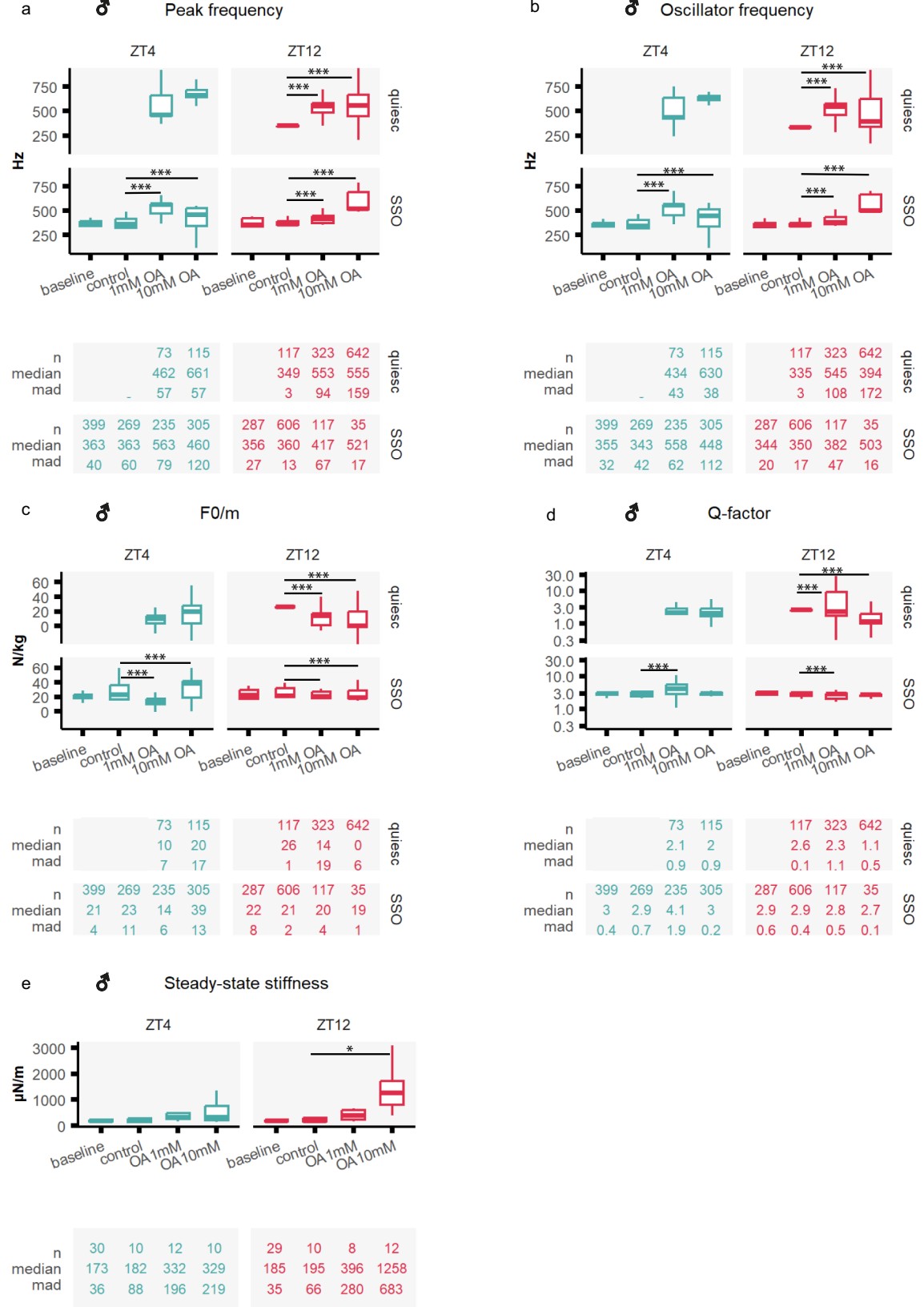

[ZT12] OA1mM: 489 ± 155.9 Hz, $p < 0.001$,). Flagellar stiffness values extracted from force-step stimulation slightly increased between control and octopamine injections at ZT4 (Fig. 5e, Supplementary tables 7–10, steady-state stiffness [ZT4] control: 93 ± 18 µN/m; [ZT4] OA1mM: 116 ± 21 µN/m, $p$.adj <0.05).

## The octopamine receptors AgOctβ2 and AgOAMB mediate novel auditory roles in malaria mosquitoes

To gain insight into octopamine auditory function, we generated knock-out mosquitoes of two octopamine receptors expressed in the mosquito ear. We selected the highest expressed octopamine

**Fig. 4 | Octopamine injection causes acute changes in auditory responses of male malaria mosquitoes.** Auditory tests were conducted after injecting three different solutions: control ringer and two different octopamine concentrations (1 mM and 10 mM). Blue boxplots (left) show ZT4 data, and red (right) ZT12. **a–d** Biophysical parameters were extracted from fitting damped harmonic oscillator functions to the flagellar responses to frequency-modulated sweeps, including the peak frequency, the oscillator frequency, the acceleration (F0/m) and Q-factor (see Methods). Flagella were assigned to either displaying a quiescent state or SSOs. No quiescent mosquitoes were observed at the baseline, neither at ZT4 nor at ZT12, nor ZT4 in control conditions, so plots are empty for these categories. Octopamine injections affected most biophysical parameters, and remarkably, caused a sharp increase in the peak and oscillator frequency. **e** Steady-state stiffness values extracted from force-step stimulation responses. Octopamine injections induced a remarkable increase in flagellar stiffness values that was more pronounced during swarm time ZT12. Central line, median; box limits, first and third quartiles; lower and upper whiskers, 5th and 95th percentiles, respectively. Tables underneath each graph display the n numbers (total number of runs (stimulated sections) that passed the curation process and are included in the analysis), median and median absolute deviation (mad) for each of the categories shown in the graph above. Significant differences between injection effects starred (two-sided Wilcoxon rank-sum tests with Holms procedure for multiple comparison correction, *$p$.adj <0.05; **$p$.adj <0.01; ***$p$.adj <0.001). OA octopamine, SSO self-sustained oscillations, ZT zeitgeber time. Sample sizes: ZT4 control = 8; ZT4 OA1mM = 8; ZT4 OA10mM = 7; ZT12 control = 11; ZT12 OA1mM = 9; ZT12 OA10mM = 11.

receptor, *AgOctβ2* (AGAP002886), and the receptor showing a peak of expression during swarm time, *AgOAMB* (AGAP000045). We used CRISPR/Cas9 to disrupt the 1st coding exon of *AgOAMB* and 3rd coding exon of *AgOctβ2* – regions of predicted octopamine binding or GPCR activity highly conserved across mosquito strains[31,34,35]. A GFP fluorescent marker was inserted into the target site, which allowed for tracking mutant alleles (Fig. 6a–b). Both mosquito mutant lines were tested for unstimulated and stimulated auditory behaviours.

**Octopamine receptor mutants presented severe defects in the pattern of fibrillae erection.** We first observed the fibrillae erection pattern of the octopamine receptor mutants compared to wildtype animals (Fig.6c). As shown above, 100% of wildtype males erected their fibrillae during the laboratory swarm time. Almost 90% of *AgOAMB* homozygous mutant males did so, too. By contrast, none of the *AgOctβ2* knock-out animals erected their fibrillae during the laboratory swarm time. We then performed 1 mM octopamine injections after mounting the mosquitoes (mounting the mosquitoes causes fibrillae collapse). Octopamine injections caused the partial or full erection of the antennal fibrillae in almost 90% of wildtype and 50% of *AgOAMB* knock-out males. However, it did not cause any erection in *AgOctβ2* mutants (Fig. 6c).

**Octopamine receptor mutants maintained SSOs upon injecting octopamine.** We then studied the flagellar mechanical state in mutant animals during the laboratory swarm time. At the baseline, all genotypes presented a similar distribution, with most animals either presenting SSOs or a transient state (Fig. 6d). Although in wildtype animals octopamine prompts a shift to quiescent states, this was not the case for any of the mutants that maintained SSOs or transient states despite octopamine injections. Regarding the SSO frequency at swarm time (Fig. 6e), *AgOctβ2* mutants exhibited higher baseline values compared to wildtype and *AgOAMB⁻* animals ([ZT12] baseline; wildtype: 359 ± 11 Hz; *AgOAMB⁻*: 353 ± 8 Hz; *AgOctβ2*: 372 ± 11 Hz, *AgOctβ2* compared to other genotypes $p$.adj <0.001). Injecting octopamine caused a shift to higher oscillator frequencies of SSO in both wildtype and *AgOAMB⁻* animals, but the effect was stronger in the mutant line. By contrast, *AgOctβ2* mutants reduce the SSO frequency upon octopamine injections (Fig. 6e, Supplementary tables 11–12, [ZT12] SSO frequency OA1mM; wildtype: 398 ± 53 Hz; *AgOAMB⁻*: 429 ± 112 Hz; *AgOctβ2*: 344 ± 16 Hz, both mutants to wildtype $p$.adj <0.001). The SSO amplitude was higher at the baseline for both octopamine receptor mutants compared to wildtype mosquitoes (Fig. 6e, [ZT12] SSO amplitude baseline; wildtype: 387 ± 257 nm; *AgOAMB⁻*: 794 ± 287 nm; *AgOctβ2*: 566 ± 430 nm, both mutants to wildtype $p$.adj <0.001). Although octopamine did not induce changes in the SSO amplitude in wildtype mosquitoes, it did affect them slightly in both mutants ([ZT12] SSO amplitude OA1mM; wildtype: 391 ± 224 nm; *AgOAMB⁻*: 623 ± 466 nm; *AgOctβ2*: 728 ± 501 nm, both mutants [ZT12] compare to [ZT4] $p$.adj <0.001).

**Octopamine did not increase the flagellar stiffness of octopamine receptor mutants.** We subjected the octopamine receptor knock-outs to frequency-modulated sweep stimulation. Data are shown for SSO

animals, as few *AgOAMB* mutants and none of the *AgOctβ2* mutants exhibited quiescent flagellar states. At the baseline, mutant mosquitos exhibited higher peak and oscillator frequencies than the wildtype line (Fig.7a–d, Supplementary tables 13–14, peak frequency [ZT12] baseline; wildtype: 356 ± 27 Hz; *AgOAMB⁻*: 380 ± 14 Hz; *AgOctβ2⁻*: 410 ± 33 Hz, both mutants to wildtype $p$.adj <0.001). Injecting 1 mM octopamine caused distinct effects in both mutant lines. The shift in all parameters analysed was more extreme in *AgOAMB* mutants compared to wildtype animals (Fig.7a–d, e.g. peak frequency [ZT12] OA1mM; wildtype: 417 ± 67 Hz; *AgOAMB⁻*: 470 ± 90 Hz, $p$.adj <0.001). By contrast, there were barely any changes in the parameters' values in *AgOctβ2* mutants.

Steady-state stiffness values were also extracted from force-step stimulation analyses (Fig. 7e, Supplementary tables 15–16). At the baseline, *AgOAMB* and *AgOctβ2* mutants presented similar steady-state stiffness values to the wild type. The impressive increase in stiffness upon octopamine injections shown by wildtype mosquitoes disappeared in the mutants; the mutants did not significantly change their stiffness values upon octopamine injections (steady-state stiffness [ZT12] baseline; *AgOAMB⁻*: 160 ± 35 μN/m; [ZT12] OA1mM: 178 ± 60 μN/m, $p$.adj > 0.05; [ZT12] baseline; *AgOctβ2⁻*: 151 ± 16 μN/m; [ZT12] OA1mM: 137 ± 37 μN/m, $p$.adj >0.05).

**The AgOctβ2-mediated octopaminergic signalling in the mosquito ear can be targeted for mosquito control**
Octopamine receptors are promising insecticide targets as they are found exclusively in invertebrates. The insecticide amitraz is an agonist of octopamine receptors[36] and is widely used as a pesticide[37–39]. Its potential to control mosquito populations has been scarcely explored[40–42] and it has never been used to target a sensory system. To investigate whether octopaminergic signalling in the mosquito ear is affected by amitraz, we tested its effects on antennal fibrillae erection. Amitraz activates both α- and β-adrenergic-like octopamine receptors in other insects[43,44], so we would expect it to induce the erection of the antennal fibrillae through activating AgOctβ2.

We exposed male mosquitoes to five minutes of 0.025%, 0.1% and 0.4% amitraz at ZT4 – when the antennal fibrillae are collapsed – and quantified the proportion of males with erected fibrillae in a time series (Fig. 8a). Five minutes after exposing the mosquitoes to any amitraz concentration, 98% of male mosquitoes erected their fibrillae. We also exposed *AgOctβ2* mutant males to 0.1% amitraz to test if this receptor was necessary for the amitraz induction of fibrillae erection. As expected, we observed that mutant mosquitoes did not erect the fibrillae following amitraz exposure (Fig. 8b), showing that AgOctβ2 is required for the physiological effects of amitraz exposure.

## Discussion
Mosquito audition mediates the recognition of mating partners within crepuscular swarms. The transient nature of swarms suggests adaptive, yet transient, modulations of hearing. Here, we found evidence of an extensive auditory neuromodulatory network in malaria mosquitoes. Octopamine receptor temporal expression peaked at swarm time, suggesting regulatory roles in the audition. To confirm this, we

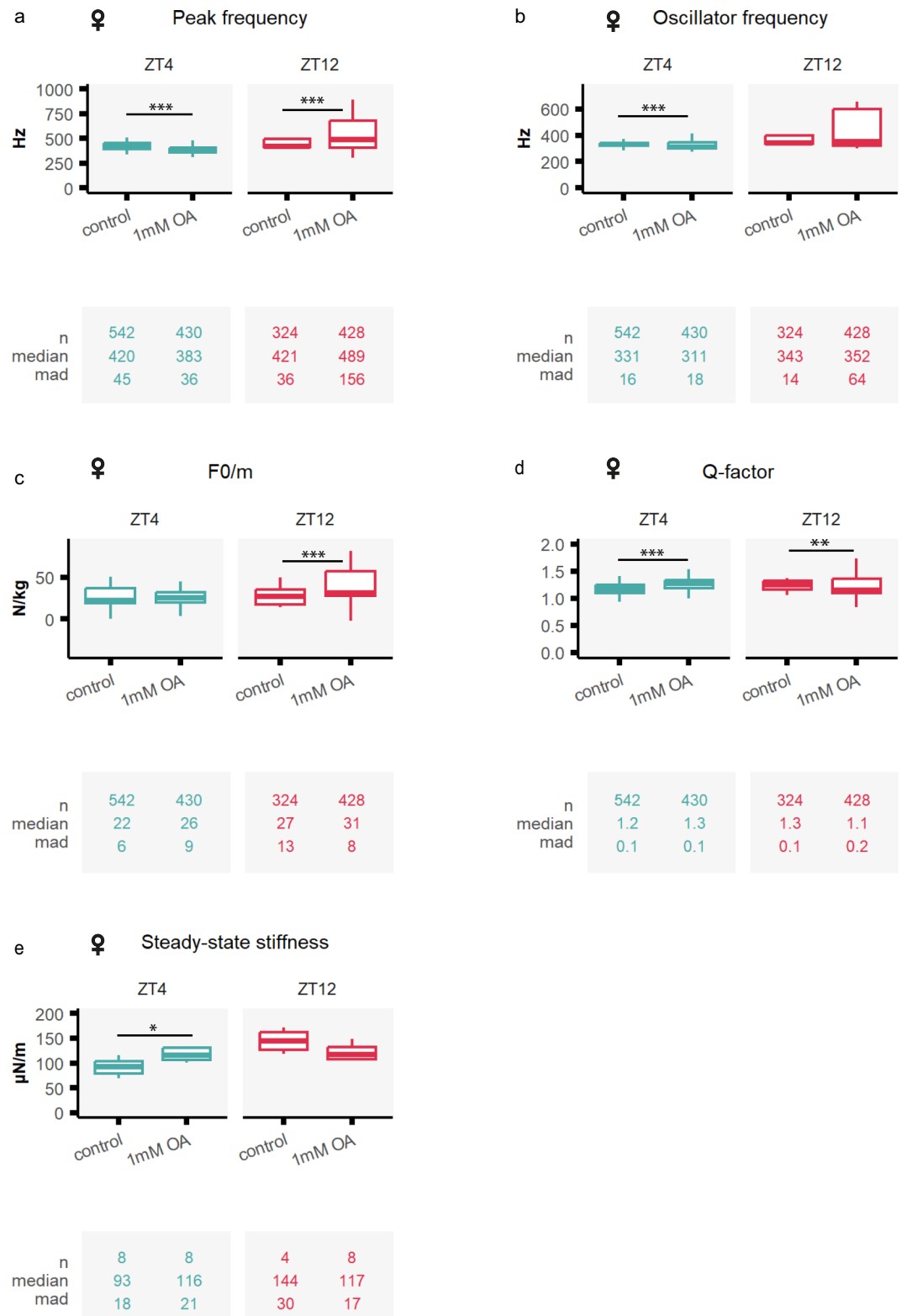

**Fig. 5 | Octopamine injection causes mild effect in female malaria mosquitoes.**
Tests of auditory biophysics were conducted after injecting two different solutions: control ringer and 1 mM octopamine concentrations. Blue boxplots (left) show ZT4 data, and red (right) ZT12. (**a–d**) Biophysical parameters were extracted from fitting damped harmonic oscillator functions to the flagellar responses to frequency-modulated sweeps, including peak and oscillator frequencies, the acceleration (F0/m) and the Q-factor. Octopamine injection had mild effects on female mosquito hearing compared to males. **e** Steady-state stiffness values extracted from force-step stimulation responses. Central line, median; box limits, first and third quartiles; lower and upper whiskers, 5th and 95th percentiles,

respectively. The tables underneath each graph display the n numbers (total number of runs (stimulated sections) that passed the curation process and are included in the analysis), median and median absolute deviation (mad) for each of the categories shown in the graph above. Octopamine injection caused a small increase in flagellar stiffness at ZT4. Significant differences between injection effects starred (two-sided Wilcoxon rank-sum tests with Holms procedure for multiple comparison correction, *$p$.adj <0.05; **$p$.adj <0.01; ***$p$.adj <0.001). OA octopamine, ZT zeitgeber time. Sample sizes: ZT4 control = 8; ZT4 OA1mM = 7; ZT12 control = 5; ZT12 OA1mM OA = 8.

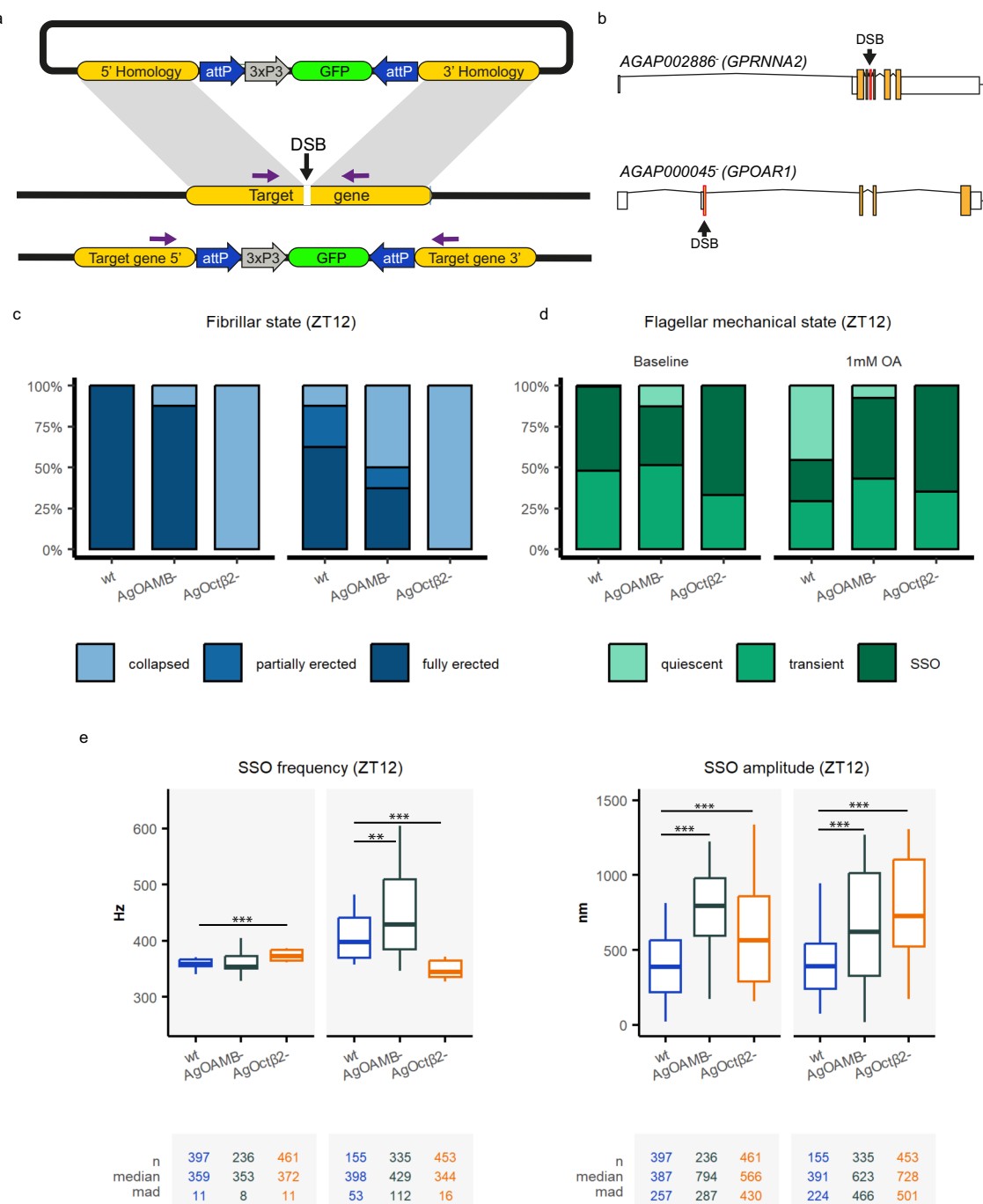

**Fig. 6 | Auditory behaviour of unstimulated ears in octopamine receptor mutants. a** Schematic of the knock-out generation approach. A double-strand break (DSB) was introduced into specific locations of the target gene using CRISPR-Cas9 mediated cleavage. A cassette containing an attP-flanked 3xP3::GFP marker construct was inserted into the target gene DSB site by homology directed repair. GFP expression was used to screen for transformants. **b** The DSB site was located in the third exon of the receptor *AgOctβ2* and the first exon of the receptor *AgOAMB*. **c** Fibrillae erection state in octopamine receptor mutants compared to wildtypes. Baseline levels before mounting the mosquitoes, show that none of the *AgOctβ2* knock-out mosquitoes erected their fibrillae during the laboratory swarm time, while almost all wildtype and *AgOAMB* mosquitoes erected them. Upon octopamine injections, *AgOctβ2⁻* mosquitoes kept the fibrillae collapse, while almost all wildtype and half of *AgOAMB⁻* animals erected them. **d** Flagellar mechanical state based on the analysis of unstimulated free fluctuations of the flagellum during swarm time. At the baseline, almost all mosquitoes exhibited exclusively SSO (except for a few *AgOAMB* mutants). Injecting octopamine pushed around half of

the wildtype flagella to a quiescent state, while the mutants did not change the flagellar mechanical state and maintained the SSOs. **e** SSO frequencies and displacement amplitudes in the different genotypes during swarm time (ZT12). Baseline levels showed an increase in SSO frequency in *AgOctβ2* mutants, but these mutants decreased the oscillator frequency upon octopamine injection. *AgOAMB⁻* animals responded with a sharper increase in the SSO frequency values than wildtypes after octopamine injections. Central line, median; box limits, first and third quartiles; lower and upper whiskers, 5th and 95th percentiles, respectively. The tables underneath each graph display the n numbers (total number of runs (unstimulated sections) that passed the curation process and are included in the analysis), median and median absolute deviation (mad) for each of the categories shown in the graph above. Significant differences between genotypes starred (two-sided Wilcoxon rank-sum tests with Holms procedure for multiple comparison correction,*$p$.adj <0.05; **$p$.adj <0.01; ***$p$.adj <0.001). DSB double-stand break, OA octopamine, SSO self-sustained oscillations, wt wildtype, ZT zeitgeber time. Sample sizes: ZT12 wildtype = 9; ZT12 *AgOAMB⁻* = 8; ZT12 *AgOctβ2⁻* = 7.

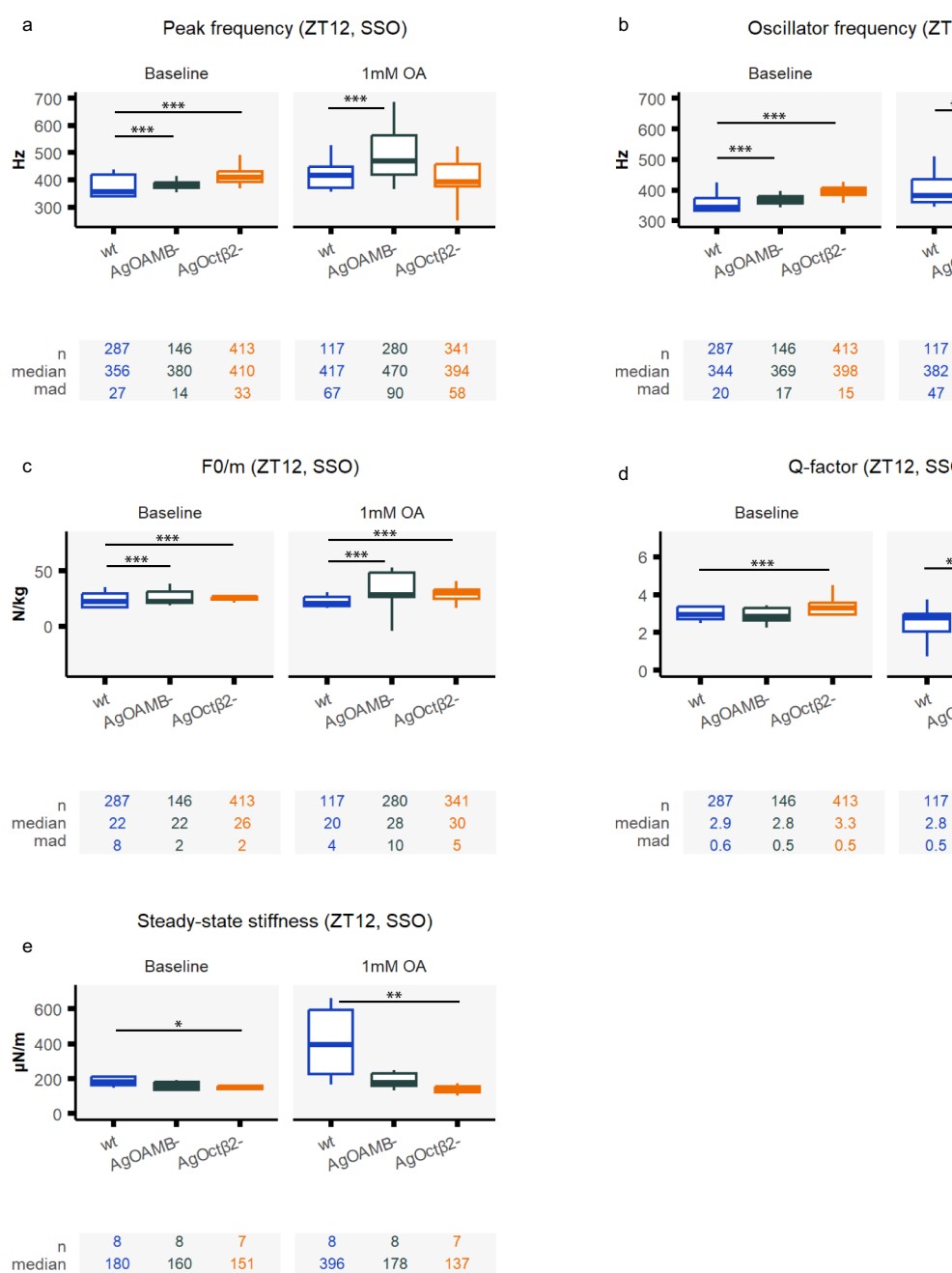

**Fig. 7 | Different phenotypes in the auditory stimulated responses of octopamine receptor mutants upon injecting octopamine.** Auditory tests were conducted after injecting two different solutions: control ringer and 1 mM octopamine concentrations. Auditory tests were only performed at ZT12 (**a–d**) Biophysical parameters were extracted from fitting damped harmonic oscillator functions to the flagellar responses to frequency-modulated sweeps, including the peak frequency, the oscillator frequency, the acceleration (F0/m), and Q-factor. As most mutant flagella were constantly exhibiting SSO, only SSO values are shown. Injecting octopamine affected most biophysical parameters in *AgOAMB* mutants, but the effects on *AgOctβ2* knock-outs were very mild. Interestingly, the changes in values in *AgOAMB* mutants were stronger than in wildtype animals (we are only showing results for SSO mosquitoes). The increase in the oscillator frequency of *AgOAMB* mutants reached values of up to 450 Hz. **e** Steady-state stiffness values

extracted from force-step stimulation responses. The sharp increase in the stiffness values observed in wildtype mosquitoes was absent in octopamine receptor mutants. In *AgOctβ2* knock-outs the stiffness values even decreased upon octopamine injections. Central line, median; box limits, first and third quartiles; lower and upper whiskers, 5th and 95th percentiles, respectively. Tables underneath each graph display the n numbers (total number of runs (stimulated sections) that passed the curation process and are included in the analysis for each category), median and median absolute deviation (mad) for each of the categories shown in the graph above. Significant differences between genotypes starred (two-sided Wilcoxon rank-sum tests with Holms procedure for multiple comparison correction, *$p$.adj <0.05; **$p$.adj <0.01; ***$p$.adj <0.001). OA octopamine, SSO self-sustained oscillations, wt wildtype, ZT zeitgeber time. Sample sizes: ZT12 wildtype = 9; ZT12 *AgOAMB⁻* = 8; ZT12 *AgOctβ2⁻* = 7.

examined the role of octopamine and demonstrated that it plays a sexually dimorphic role and modulates mostly male hearing. We showed that octopamine affects different auditory parameters that determine the male's ability to detect female flight tones; modulated

system properties included auditory tuning and flagellar stiffness. We identified an α-adrenergic-like (*AgOAMB* or *AGAP000045*) and a β-adrenergic-like (*AgOctβ2* or *AGAP002886*) octopamine receptor mediating octopamine's auditory roles. Individual gene knock-out of both

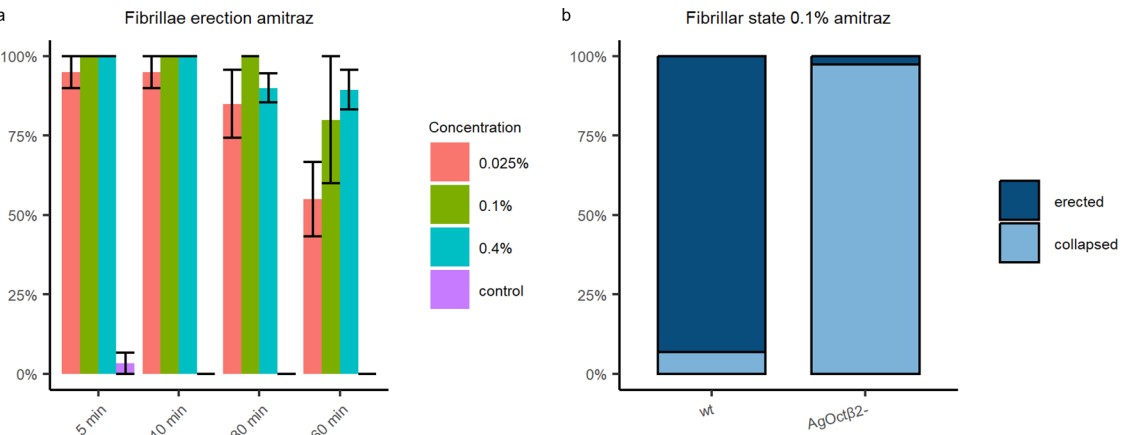

**Fig. 8 | Amitraz targets the octopaminergic signalling in the malaria mosquito ear. a** Time series of the effects of different amitraz concentrations on the fibrillae erection of malaria mosquitoes. Almost 100 % of male mosquitoes erect their fibrillae upon amitraz exposure. Data are presented as mean values +/− SEM. **b** *AgOctβ2* knockouts exposed to amitraz do not erect their fibrillae. Concentrations used based on[89]. Samples sizes: time series experiments: control = 30; 0.025% amitraz = 10; 0.1% amitraz = 5; 0.4% amitraz = 20; amitraz exposure *wt* = 29; *AgOctβ2* knockouts = 39.

receptors resulted in different auditory phenotypes and critically, *AgOctβ2* mutants fail to erect their antennal fibrillae. We also demonstrated that the octopaminergic pathways in the mosquito ear can be targeted by currently available insecticides, opening new pathways for mosquito control.

The broad repertoire of neurotransmitters and neuropeptide receptors identified in our RNA-sequencing analysis suggests a high plasticity of mosquito hearing. Immunostaining and behavioural confirmation of these findings will help to interpret these results. We hypothesize that this sensory plasticity enables the dynamic adaptation of the auditory function to environmental changes[45–48] or variations in the swarm's sensory ecology (related e.g. to mosquito numbers or noise levels) to facilitate female detection. As the male mosquito ear harbours more than 16,000 sensory neurons, neuromodulatory innervation could also contribute to establishing neuronal subtypes with different sensitivity to sound and adaptation capabilities[49]. It is also plausible that the neuromodulators target support cells in the JO to affect the mechanotransduction process[50–52]. Investigating the distribution and function of the neuromodulatory input seems essential to understand the outstanding performance of the mosquito ear as a sound detector.

Our transcriptomic analyses suggest that, as previously described in *Cx. quinquefasciatus* mosquitoes[7], the biogenic amines octopamine and serotonin also innervate the malaria mosquito ear, as we detected the expression of several octopamine and serotonin receptors. In insects, biogenic amines modulate various aspects of sensory organ physiology[16]. In particular, serotonin modulates mating behaviour[53] and the circadian clock entrainment to visual stimuli[54]. Our transcriptomic data support the role of serotonin *in An. gambiae* audition, as was recently shown for *Ae. aegypti*[55]. Several GABA-A, GABA-B, nicotinic acetylcholine, and neuropeptide receptors were also identified, some with known auditory functions in vertebrates[51,52], suggesting a complex modulation of mosquito auditory physiology.

Male malaria mosquitoes are attracted to female sounds[56,57] preferably during swarm time[9]. To investigate the molecular nature of this circadian-time dependency of auditory responses, we interrogated our transcriptomic dataset of neurotransmitter and neuropeptide receptors for those showing cycling expression. The single outcome in males was the octopamine receptor *AgOAMB*, hinting at a role of octopamine in mosquito audition during swarming (Fig. 1). Octopamine had already been implicated in mosquito auditory modulation at swarm time as it induces fibrillae erection in *An. stephensi*[10], a phenomenon temporarily limited to the swarming period. We demonstrated that, also in *An. gambiae*, octopamine controls the erection of the antennal fibrillae (Fig. 3a). We also detected an effect of octopamine in modulating the flagellar mechanical state by promoting quiescent states over SSOs (Fig. 3b). SSOs have been linked to the amplification of female flight tones in the swarm[5,6,8] It seems therefore that octopamine acts on two competing frontiers: one that modulates the SSO to higher frequencies which favour female audibility[58] while at the same time promoting the opposing mechanical state of the flagellum into quiescence. We could infer that there is a fine balance based on the total concentration of octopamine within the system. We should also consider that our knowledge of SSO mechanisms and functions is still incomplete. The SSOs seen in the flagellar ears of male *Anopheles* mosquitoes are unique across auditory systems, both in magnitude and properties[8]. The fact that their energy content varies by several orders of magnitude across different mosquito species[8] also points to substantial degrees of ecological diversification. It seems beyond doubt that SSOs are a crucial component of the mosquito hearing mechanism, but it will require further studies dedicated to SSOs to explore them on all relevant functional levels (e.g. within the entire plane of flagellar mobility and binaurally, across both ears) and to finally understand how they support the male ear in the detection of faint female flight tones. Here, circadian modulations mediated by octopamine through different, functionally distinct, receptors clearly play a vital role but there are still many unknowns. We would argue that in our experimental setup, octopamine effects on the mechanical state are too severe compared to its natural role in the auditory system, where it probably just modulates certain SSO parameters (e.g. frequency and amplitude). In our experiments, injected octopamine reaches the mosquito ear through the haemolymph rather than via efferent terminals, activating all octopamine receptors at once. Therefore, the spatial and time resolution of the system is altered, potentially inducing the cessation of SSOs rather than a modulation of their properties. Moreover, gluing the mosquito ear for our auditory tests might induce stress responses that can affect the flagellum, and SSOs (e.g[59]). Our results are consistent with the canonical view that the active processes found to support hearing across taxa (such as e.g. SSOs in mosquitoes) only operate within a certain space of biophysical values and show critical dependence on the state of distinct, yet mostly unidentified, control parameters; octopamine may very well be one of those. We suggest that although injected octopamine induces SSO cessation this is not necessarily what occurs in nature. Notably, in the bullfrog sacculus, increasing $Ca^{+2}$ levels modulate spontaneous oscillations of the hair bundle by increasing its frequency and reducing its amplitude until they are suppressed[60], the same effect that we observe for octopamine in mosquito flagellar oscillations. The α-octopamine

receptor AgOAMB signals through an increase in intracellular Ca$^{+2}$[12], suggesting parallel modulatory mechanisms of vertebrate and mosquito spontaneous oscillations of their auditory system despite obvious structural differences.

Putting together the available knowledge on how acoustic detection works in swarms of malaria mosquitoes, we suggest that the main role of octopamine is to modulate SSO performance. Current theories support that the male detects the female wingbeat frequency using a distortion product (DP) system: low frequency DPs are generated in the male ear by the mixing of the male and female wingbeats[58]; males listen out for these distortions instead of the actual flight tones of females. Recent research shows that males increase their wingbeat frequencies at swarm time to enhance the audibility of females via the DPs[58]. SSOs, which increase flagellar displacement responses to female wingbeat frequencies, would thus support the male in detecting the female by increasing the amplitude of the DPs generated. The daily modulations of male wingbeat frequency, which also include a marked increase during swarm time[58], might be part of more complex audibility adjustments, which also necessitate SSO frequency changes to maximize DP production in the male flagellum and boost female audibility. In line with these assumptions, our data show that octopamine injection induces an increase in SSO frequency (Fig. 3c). We speculate that by modulating the SSO frequency, octopamine might contribute to adapting the male auditory system to optimize female detection. Interestingly, octopamine also innervates flight muscles[61] and modulates flight performance[11]. It is plausible that octopamine plays a multifunctional role in modulating the female acoustic detection system in the swarm: it induces the erection of antennal fibrillae, tunes the auditory system, and potentially modulates the wingbeat frequency. More experiments and mathematical modelling are required to test and probe this hypothesis.

Injecting octopamine also increases the flagellar steady-state stiffness (Fig. 4e). This could partly explain the observed effects on the other biophysical parameters. The flagellar ears of mosquitoes, especially those of males, are complex oscillators and the interrelations of their functional parameters are not well understood. Neuropharmacological manipulations, such as those presented here, will be a vital tool to dissect the underlying complexity. Octopamine terminals have been observed at the base of auditory cilia in *Cx. quinquefasciatus* mosquitoes[7]. If adaptation in the mosquito flagellar ear is complete – as has been shown for *Drosophila* before – then the flagellar steady-state stiffness represents the combined elasticity of those components that suspend the sound receiver, but which do not directly contribute to the transmission of the sound stimulus to the sensory transducer channels proper[62,63]. The location of octopamine terminals at the auditory cilia base could indicate that octopamine stiffens the flagellum by increasing the ciliary tension.

The role of female hearing in mosquito partner detection remains elusive. Only males are attracted to female sounds[9,56,57]. There are very few examples of sound-induced behavioural responses in females[64]. Our transcriptomic analysis showed nonetheless that most receptors identified in males were also expressed in females. As the efferent pattern of *An. gambiae* female JO is limited[8], the ligands of the receptors identified in the RNA-Seq analysis might reach the ear through the haemolymph acting as neurohormones. As shown here, octopamine auditory effects are highly sexually dimorphic and females respond very faintly to octopamine injections (Fig. 5). It would be important to validate if the comparisons showing statistical significance imply biological relevance. Given the minute nature of some of the effects observed, we cannot assume their biological relevance until demonstrated by further studies. The sexual dimorphism in octopamine effects agrees with previous studies that support a predominant role of the male in the acoustic detection of the mating partner[58,65]. However, both the transcriptomic analysis in this paper and previous anatomical studies[8] support that the female ear is

complex. New approaches are needed to elucidate the role of female hearing during mating, as the tools utilized to study male hearing have not yielded new insights. Developing finer behavioural assays might be necessary to understand if and how female audition contributes to mating partner detection.

In this study, we identified the receptors mediating octopamine's auditory role. AgOctβ2 is a β-adrenergic-like octopamine receptor orthologue of Octb2R in other insects[66]. Most octopamine-induced auditory responses observed, including the erection of the antennal fibrillae and the flagellar mechanical effects, were nearly abolished in *AgOctβ2*– mutants, supporting its role as the primary octopamine receptor in the mosquito ear (Figs. 6, 7). The other octopamine receptor detected, the α-adrenergic-like octopamine receptor *AgOAMB*, showed a peak of expression at swarm time, which probably accounts for the stronger effects of octopamine at swarm time for most parameters studied. *AgOAMB* mutants displayed an intermediate phenotype, with octopamine-mediated auditory effects being reduced compared to wildtype animals, both regarding the fibrillae erection and the flagellar stiffness. Interestingly, none of the receptor mutants ceased SSOs upon octopamine injections (Fig. 6d), supporting our previous argument that, in nature, octopamine would modulate SSOs rather than stop them if, for example, not both receptors are activated at the same time by the octopaminergic efferent innervation.

Also, the auditory responses of *AgOAMB* mutants can be misleading and need to be discussed. The fact that some biophysical parameters extracted from frequency-modulated sweep stimulation show stronger value shifts upon octopamine injection in *AgOAMB* mutants compared to wildtypes (e.g. peak and oscillator frequencies, Fig. 7) does not mean that octopamine has a stronger effect on *AgOAMB* mutants. Instead, the octopamine effect is stronger in wildtypes and therefore it pushes flagella to quiescent states, so they are removed from the dataset in the graph (Figs. 6, 7) that shows only SSO responses. In *AgOAMB* mutants, because the stiffness shift is weaker, mosquitoes can maintain SSOs, and this is reflected in stronger shifts in the other biophysical parameters that are nonetheless not powerful enough to induce SSO cessation, as it occurs in wild types. In *AgOctβ2* mutant mosquitoes, SSO values barely changed after octopamine injection. Moreover, our data support that the function of these receptors is non redundant and complementary, as knocking out either receptor shows distinct auditory phenotypes. They also support some constitutively signalling of the AgOctβ2 receptor during swarm time, as baseline differences in some auditory parameters are present when compared to wildtypes. It is plausible that AgOctβ2 plays a more fundamental role in preparing the male ear for swarm time, while AgOAMB modulates finer adjustment of the mechanical tuning to adapt the ear to the distinct sensory ecology of the specific swarm where the male mosquito is trying to find a female.

The fact that β- adrenergic-like (AgOctβ2) and α- adrenergic-like (AgOAMB) octopamine receptors use different second messengers, namely an increase in Ca$^{2+}$ and an increase in cAMP[12], respectively, further supports a functional distinct role of both octopamine receptors. These two octopamine receptors are also found in mechanosensory neurons of the spider *Cupiennius salei* that are innervated by efferent octopaminergic fibres[67]. In the spider mechanosensory neurons, each receptor modulates different aspects of neuronal sensitivity, including baseline levels and rapid changes[68]. Further experiments are necessary to disentangle the specific auditory role of each receptor in mosquitoes, and factors to consider include whether they localize to different neuronal regions, to different cells (e.g. different neurons in the scolopidia or support cells) or if they are involved in temporal changes in the auditory physiology.

We lastly tested whether the octopaminergic signalling in the mosquito ear could be targeted using pharmacological interventions to potentially interrupt mosquito partner recognition by disrupting mosquito hearing and thus eventually help control mosquito

populations. We used the insecticide amitraz, an octopamine receptor agonist, as a proof-of-concept of this approach. Amitraz is widely used as a pesticide to kill ticks and parasitic mites[37,38], however, it is not used as an insecticide against mosquitoes probably due to its low toxicity[69]. We found that amitraz exposure activates AgOctβ2 in the mosquito ear and induces the erection of the antennal fibrillae. It would be interesting to explore if amitraz, or preferably another compound acting as an octopamine receptor antagonist -to mimic the auditory effects observed in *AgOctβ2* knockout mosquitoes-, would disrupt mosquito mating by impairing the capacity of male mosquitoes to detect, or tune into, female flight tones. Such *mating disruptors* are promising novel tools for mosquito control[70,71] that could be delivered in the field using sugar baits[72]. This would be of the highest priority as the emergence of insecticide resistance and behavioural adaptations across malaria mosquitoes is endangering malaria control[73]. Mating disruptors have been largely used for the control of insect pests[74]. To implement them for mosquito control, more research is undoubtedly required to disentangle the neurological pathways involved in mosquito mating, identify suitable molecules to disrupt them and develop better delivery methods to target male mosquitoes. The exquisitely complex mosquito auditory system provides numerous opportunities to explore the potential of applying mating disruptors for mosquito control.

## Methods

### Mosquito rearing and entrainment

*An. gambiae* Kisumu strain mosquitoes used for the RNA-Seq analysis were provided by Shahida Begum from the London School of Hygiene and Tropical Medicine. All other experiments used *An. gambiae* G3 strain mosquitoes, initially provided by Andrea Crisanti (Imperial College London)[75] until an independent colony was established at the UCL Ear Institute.

Experimental pupae were sex-separated and males and females were hatched in different cages. Mosquitoes were then transferred to environmental incubators (Percival I-30 VL Multipurpose Plant Breeding Chamber, CLF PlantClimatics GmbH, Germany) for circadian entrainment. Entrainment conditions consisted of a 12 h: 12 h light/dark cycle (light intensity during the light cycle was 80 μmolm-2 s-1 from 4 fluorescent lamps) with 1-hour ramped light transition to simulate dawn and dusk (ZT0-ZT1 and ZT12-ZT13, respectively; ZTX is the formalized notation of an entrained circadian cycle's phase). Temperature and relative humidity were set constant throughout the circadian day, at 28 °C and 80% respectively. Mosquitoes were exposed to circadian entrainment in incubators for three days before performing any experiment. All experiments were conducted with 3-to-7-day old mosquitoes.

### RNA-Seq

**Sample collection and preparation.** On day four of circadian entrainment, 35 male and female mosquitoes were removed from the incubators at six different circadian time points (ZT0, ZT4, ZT8, ZT12 or swarm time, ZT16, ZT20), sedated with $CO_2$ and transferred to Eppendorf tubes, which were in turn immediately transferred to liquid nitrogen to minimize sample degradation. The samples were kept in a storage freezer at −80 °C prior to dissections. During dissections, using a pair of forceps, flagella were first dissected off the JOs, which were subsequently removed from the mosquito heads and placed in Eppendorf tubes, immersed in ice. The mosquito mouthparts were also removed and the remainder of the heads were too transferred to (separate) Eppendorf tubes.

**Library preparation.** RNA was extracted (Quiagen) and its integrity was confirmed using Agilent's 2200 Tapestation.

The cDNA libraries were prepared by UCL genomics. Samples were processed using the SMART-Seq v4 Ultra Low Input RNA Kit (Clontech

Laboratories, Inc.). Briefly, cDNA libraries were generated using the SMART (Switching Mechanism at 5′ End of RNA Template) technology which produces full-length PCR amplified cDNA starting from small amounts (500 pg) of total RNA. 9 cycles of PCR were used to generate cDNA. The amplified cDNA was checked for integrity and quantity on the Agilent Bioanalyser using the High Sensitivity DNA kit.

150 pg of cDNA was then converted to a sequencing library using the Nextera XT DNA protocol (Illumina, San Diego, US). This uses a transposon able to fragment and tag double-stranded cDNA (tagmentation), followed by a limited PCR reaction (12 cycles) which adds sample specific indexes to allow multiplex sequencing.

**Sequencing.** Libraries to be multiplexed in the same run were pooled in equimolar quantities, calculated from Qubit and Tapestation fragment analysis.

Samples were sequenced on the NextSeq 500 instrument (Illumina, San Diego, US) using a 38 paired-end run, generating 400 M read pairs in total.

**Data analysis.** Figure 1b shows a schematic of the data analysis workflow. FastQC[76] and MultiQC[77] were used to perform quality control of sequencing reads. These were subsequently classified, and read counts were estimated, with Kallisto[78], based on the *Anopheles gambiae* transcriptome (https://vectorbase.org/vectorbase/app/downloads/release-53/AgambiaePEST/fasta/data/VectorBase-53_AgambiaePEST_AnnotatedTranscripts.fasta). Following a preliminary principal component analysis of sample counts, a male JO sample collected at ZT12 was deemed unsuitable for use in further analysis and discarded.

1. Identifying transcript expression counts above the quantification noise floor

   The remaining samples were used in classifying what we hereby call the male and female JO *expressed* genes. The idea behind this analysis was to identify the noise distribution of estimated transcript counts within the context of our RNA-sequencing experiment: A quantification of falsely and randomly assigned read counts. Contrasted to this distribution, then, we could make statistical claims of whether transcripts' estimated expressions (a) exceed the count noise floor and can thus be regarded as biologically expressed, or (b) do not exceed the count noise floor and thus be regarded as noise. The formulation of the noise distribution was conducted as follows:

   First, 50 transcripts were randomly selected, noted, and removed from the *Anopheles gambiae* transcriptome. Then, the remaining transcripts were used to simulate paired end reads with lengths and coverage similar to our sequenced reads. Simulation of reads was conducted with NEAT[79]. The 50 transcripts that had been initially removed were subsequently re-introduced into the dataset to recover the whole transcriptome, and read counts for the simulated reads were estimated using Kallisto. Count estimates for the originally removed transcripts were extracted. Note that any counts obtained from these genes can be regarded as noise/false assignments, as they were not part of the dataset used to simulate reads to start with. The mean expression counts for these 50 transcripts – the mean noise of counts – were calculated. The whole process was repeated ten times, each time removing a different set of randomly selected transcripts, to obtain ten means for noise counts. These means were normally distributed (Shapiro-Wilk test), and their mean was calculated, assumed to represent the average of the underlying population of count noise, and used to define a Poisson distribution for the counts' noise. This Poisson distribution served as the reference distribution against which transcript expression counts were compared.

   For each gene, mean expression counts, across all circadian time points were compared against the noise distribution.

If expression counts for that gene were high enough such that the probability of them arising solely by noise, given the noise distribution, was <0.05, then that gene was regarded as expressed. Probability values for these expression counts were examined after adjustment for multiple comparisons (Benjamini-Hochberg method).

2. Classifying transcripts by gene ontology (GO)

Transcripts identified as expressed were further investigated in search of those that were assigned with GO terms related to neurotransmission and neuromodulation (Table 1).

If any genes were assigned with a GO term 'child' ( = subterm) of any of the above, but not with the 'parental terms' themselves, then we assigned it to the parental GO ourselves. A perusal through the hierarchy of GO terms for GO inheritance was conducted with the Python package goatools[80].

GO annotations of each of the transcripts of *Anopheles gambiae* were compiled from two sources:

i. The *Anopheles gambiae* gene association file (gaf) is available for download on VectorBase (https://vectorbase.org/vectorbase/app/downloads/release-53/AgambiaePEST/gaf/VectorBase-53_AgambiaePEST_GO.gaf)

ii. GO annotations of *Anopheles gambiae* transcript orthologs. This approach was only applied to instances where the transcript had not itself been annotated, but had a close ortholog that was assigned with any of the GO terms of interest. We focused on other mosquito species and drosophilids for finding orthologous genes.

3. Differential expression analysis between male and female JOs

Normalization of read counts for comparability across, and comparison between male and female samples was conducted in R, with the package DESeq2[81].

The circadian time of sampling introduces variation in transcript abundances, but the focus of this analysis was the source of variation due to the sex factor (male vs female). To adjust the read counts accordingly, and increase the sensitivity of finding transcriptional differences specific to the sex factor, the time factor was included in the design formula (i.e. ~ time + sex).

The significance condition applied to comparisons was a False Discovery Rate (FDR) of <0.05.

4. Cycling expression analysis of male and female JOs

The cycling expression analysis was conducted individually for male and female samples. Normalization of read counts for comparability across the circadian time-points and a first round of statistical filtering of potential cycling candidates was conducted in R, with DESeq2. For the analysis of the diel pattern of gene expression (the aforementioned statistical filtering step), briefly, a "full" model of the counts was constructed including a term for time, and compared to a "reduced" model (lacking the time term) with the likelihood ratio test function of DESeq2 (conceptually similar to ANOVA) – the null hypothesis being that a model of the data incorporating time offers no improvement over one which doesn't. The significance condition applied was FDR < 0.05.

Transcripts that passed the first round of filtering were further investigated for cycling via the R-package MetaCycle[82], using the JTK_CYCLE function for rhythmicity detection. Once again, the significance condition applied was FDR < 0.05.

## CRISPR-Cas 9 mediated mutant generation

**Generation of gRNA constructs.** We designed gRNAs targeting coding regions of *AgOAMB* (AGAP000045) and *AgOctβ2* (AGAP002886) using CRISPOR (http://crispor.tefor.net/). We selected gRNAs with no SNPs reported in *An. gambiae* G3 strain (*AgOAMB* gRNA sequence: GTGGACGGATCCGACCAATC; *AgOctβ2* gRNA sequence: GGTCGTTCATGTGTGACGTG). gRNAs were cloned by Golden Gate cloning

(NEB) into a BSaI digested p165 vector (kindly donated by Roberto Galizi). The plasmid p165 contains the vasa2 promoter driving the expression of a human codon-optimized version of the *Streptococcus pyogenes* Cas9 gene (hCas9) and vasa 3' UTR regulatory sequence upstream a U6::gRNA cassette containing a spacer cloning sites, all flanked by attB recombination sites[83]. The full sequence of vector p165 has been deposited to GenBank (accession ID: KU189142).

**Generation of donor constructs.** Homology-directed repair was used to disrupt the coding sequences of the targeted genes. Homology arms were cloned by Gibson cloning into a donor vector (kindly provided by Roberto Galizi) that had been designed to contain an *attP*-flanked 3xP3::GFP marker construct enclosed within the homology arms, as well as an external *3xP3::RFP* marker. Homology arms extended 2 kb in either direction of the expected CRISPR-Cas9 cleavage site and were amplified from genomic DNA using the primers that include overlapping ends for Gibson cloning (underlined):

AgOAMB:

AgOAMB_5'_attP_F (CTCGAGTTTTTCAGCAAGATGTGTACCGCTC GAATCCAAC); AgOAMB_5'_attP_F (CCAGTTGGGGCCACTGTACGGAC GCGAG);

AgOAMB_5'_3'_F (CGTACAGTGGCCCCAACTGGGGTAACCTTT);
AgOAMB_5'_3'_R (GGCGAGCACCCCCCAACTGGGGTAACCTTT);
AgOAMB_3'_attP_F (CCAGTTGGGGGGTGCTCGCCTTCATCAAC) and AgOAMB_3'_attP_R (AGGAGATCTTCTAGAAAGATTTACTCCTCCA GACCCCGTA)

AgOctβ2:

AgOctβ2_5'_attP_F (CTCGAGTTTTTCAGCAAGATGTCGTGTGCCT GGCCTC); AgOctβ2_5'_attP_F (CCAGTTGGGGACATCCAGACGTCG ACCG);

AgOctβ2_5'_3'_F (GTCTGGATGTCCCCAACTGGGGTAACCTTT);
AgOctβ2_5'_3'_R (GAGGCCGTGCCCCCAACTGGGGTAACCTTT);
AgOctβ2_3'_attP_F (CCAGTTGGGGGCACGGCCTCGATCCTG) and AgOctβ2_3'_attP_R (AGGAGATCTTCTAGAAAGATGAGAGAGCGAGAGA GCAAGA)

**Microinjection of mosquito embryos and selection of transformants.** Freshly laid *Anopheles gambiae* G3 embryos were aligned and used for microinjections as described[84]. The donor construct (200 ng/μl) containing regions of homology to the relevant target locus was injected together with the relevant CRISPR plasmid (200 ng/μl). All surviving G0 larvae were crossed to wildtype mosquitoes and F1 positive transformants were identified using a fluorescence microscope (Eclipse TE200) as eGFP+ larvae. To confirm that candidate transformants were knock-outs of the gene of interest, a PCR was performed with primers binding at both sites of the homology arms in the genomic DNA region. Primers used were:

AgOAMB-F gcgattcgcgtcccataaac; AgOAMB-R gcgacccaatccc cctttt;

AgOctβ2-F (cgcccggggtacaatgtctta); AgOctβ2-R (ttggcacgaatgaca acagc).

## Tests of auditory function

**Laser doppler vibrometry (LDV).** Glass vials containing five mosquitoes were extracted from the environmental incubators at the required circadian time points. Mosquitoes were glued to a Teflon rod using blue-light-cured dental glue (as reported previously for both mosquitoes[8] and *Drosophila*[63]). The mosquito body was immobilised via glue application to minimise disturbances caused by mosquito movements but leaving thoracic spiracles free for the mosquito to breathe. The left flagellum was glued to the head and further glue was applied between the pedicels, with only the right flagellum remaining free to move.

Following this gluing procedure[8], the rod holding the mosquito was placed in a micromanipulator on a vibration isolation table, with the mosquito facing the laser Doppler vibrometer at a 90° angle.

To minimise mechanical disturbances, the laser was focused on the second flagellomere from the flagellum tip in males and the third flagellomere from the tip in females. All recordings were done using a PSV-400 laser Doppler vibrometer (Polytec) with an OFV-70 close up unit and a DD-500 displacement decoder. Spike 2 version 10 was used for data collection. Figure 2 shows a sketch of the laser Doppler vibrometry (LDV) experimental paradigm. All measurements were taken in a temperature-controlled room (22 °C) at the different circadian times specified in each experiment.

**Compound injection procedure.** One millimolar and 10 mM octopamine hydrochloride (Sigma-Aldrich catalogue O0250) solutions in Ringer[85] were prepared fresh on the day of the injection experiment. Sharpened micro-capillaries were filled with either octopamine or control ringer solutions. The tip of the micro-capillary was inserted into the thorax of a mounted mosquito and the solution was injected to flood the entire inset body and reach the JO (Supplementary Fig. 2). In all injection experiments, the protocol included creating recordings at three distinct stages: (1) baseline prior to any injections; (2) following injection of a control ringer solution; (3) following injection of either 1 mM octopamine, 10 mM octopamine or a 2nd ringer control injection. This protocol allowed us to collect baseline and compound injection data in the same experiment for comparative purposes, as well as ensure that the mosquito was healthy before any injections.

**Fibrillae erection assessment.** The antennal fibrillar state (collapsed vs erected) was assessed at the baseline or after the compound injection.

### LDV data analysis

All data have undergone a curation process to ensure that the laser was appropriately positioned on the mosquito flagellum during an experimental run. This is done through a diagnostic channel which measures and records the laser backscatter of the LDV. LDV data analysis was performed using Mathematica v.13.1.

**Force-step stimulation recordings and analysis.** The force-step stimulation protocol was used to calibrate the maximum flagellar displacement to approximately ±8000 nm and to extract some principal parameters of flagellar mechanics (Fig. 2c). Here, the flagellum was stimulated using electrostatic force-step stimuli. LDV measurements of flagellar displacements and electrophysiological activity were recorded simultaneously. For analysis, mosquito apparent flagellar mass estimates were calculated as in[8]. Force-step stimulation analysis was performed as in[86].

**Frequency-modulated sweep stimulation and analysis.** All experimental runs are segregated into two large categories: free fluctuation runs, where the flagellum is allowed to operate in its natural state unstimulated, and stimulus runs, where an electrostatic waveform is imposed on the flagellum to act as a driving force (Fig. 2d). The experimental structure interleaves unstimulated and stimulated periods, with unstimulated and stimulated parts continually alternating. Each 'run' contains one unstimulated and one stimulated dataset with a duration of 1 s each. The pipeline for each of those experimental runs as well as a novel formalism of the auditory mechanical state is detailed below.

**Unstimulated free-fluctuation section analysis.** The unstimulated data form the basis of the LDV analysis as the frequency, amplitude and mechanical state can be determined prior to the stimulated counterpart. The frequency and amplitude of the oscillator are extracted through a Fast Fourier transform (FFT), whereas the mechanical state is determined by looking at the amplitude distribution of the flagellar motion. The amplitude distribution of a fully quiescent flagellum

follows a standard normal distribution, whereas an SSOing animal would instead follow an arcsine distribution[87]. An added complication comes from the appearance of transient mechanical states which tend to lie somewhere in between a QUIES and SSO state. To account for those states, we generate a mixed distribution comprised of an arcsine and normal distribution, controlled by a weight parameter α which ranges between 0 (denoting purely QUIES) and 1 (denoting purely SSO):

$$\mathcal{D}(\alpha, x) = \alpha \frac{1}{\pi\sqrt{(x - A_{min})(A_{max} - x)}} + (1 - \alpha)\frac{1}{\sqrt{2\pi}\sigma}\, e^{-\frac{(x-\mu)^2}{2\sigma^2}}, \quad (1)$$

where α denotes the weight parameter, $A_{min}$ and $A_{max}$ the amplitude extremes of the input signal, $\sigma$ the width of the normal distribution and $\mu$ the centre of the distribution.

Due to the extreme variability of transient states in terms of their distribution profile, they are excluded from the majority of the LDV analysis. To permit some level of flexibility on the mechanical state, the cut-off condition for QUIES and SSO animals is set between [0–0.1] and [0.9–1.0] respectively and requires goodness of fit to account for 99.7% of the data.

We further formalise the SSO definition by adding another criterion where an animal is only classed as SSO if it sustained its oscillatory behaviour for three consecutive experimental runs, corresponding to a total span of six seconds. This condition was set to avoid attributing the SSO label to a briefly oscillating animal.

**Frequency-modulated sweep stimulation section analysis.** The stimulated half of each experimental run is performed by electrostatically driving the flagellum[8,86]. The waveform applied is a linear frequency sweep which ranges from 0 Hz to 1 kHz and its mirrored version of 1 kHz to 0 Hz (Fig.2d). Each experimental run is always followed by its mirrored version to determine any latency due to hysteresis.

The profile of the stimulated data follows a well-known line shape that is defined by the driven damped harmonic oscillator. The so-called 'envelope' of the waveform is extracted using a statistic-sensitive nonlinear iterative peak clipping (SNIP) method of background estimation. The envelope is then fitted through the theoretical expectation from the driven harmonic oscillator:

$$X(\omega) = \frac{F_0/m}{\sqrt{(\omega_0^2 - \omega^2)^2 + (2\zeta\omega_0\omega)^2}}, \quad (2)$$

where $F_0$ the driving force, m the effective mass of the flagellum, $\omega_0$ the natural frequency of the oscillator, $\zeta$ the damping ratio and $\omega$ the driving frequency. To reduce the degrees of freedom of the system for fitting purposes, the force and effective mass terms are absorbed into a singular parameter which represents the acceleration experienced by the system.

These fitted parameters convey important biophysical information which fully characterises the system. A commonly used derived parameter that describes a damped oscillator is the Q-factor, defined as $Q = 1/2\zeta$, which can be used to qualitatively interpret the level of dampening experienced by the flagellum.

Another useful derived parameter is that of the peak frequency of the driven damped oscillator $f_{opt} = \frac{f_0}{\sqrt{1 - 2\zeta^2}}$, which describes the frequency at which the oscillatory system achieves its maximum amplitude as a function of the driving frequency. In the limit case where $\zeta \to 0$, $f_{opt} = f_0$ where the maximum amplitude of the oscillator occurs at exactly the natural frequency of the oscillator. Any deviation of the damping ratio from 0 would lead to an apparent shift of the observed maximum amplitude at frequencies higher than the natural frequency of the oscillator.

## Amitraz exposure experiments

Glass vials were coated with 0.2 ml of 0.025%, 0.1% or 0.4% amitraz (N′-(2,4-Dimethylphenyl)-N-{[(2,4-dimethylphenyl)imino]methyl}-N-methylmethanimidamide, Sigma-Aldrich catalogue 45323) in acetone or acetone alone as a control. Vials were allowed to dry for two hours in a dark and cool room. Mosquitoes were collected from experimental cages at ZT4 and placed in paper cups. Single mosquitoes were carefully aspirated into the glass vials and exposed for 5 min to amitraz. Upon exposure, mosquitoes were transferred back to the paper cups and provided with a 10% sucrose solution. Fibrillae erection was assessed 5 min, 10 min, 30 min and 60 min after exposure.

## Statistical analysis

Samples sizes for LDV experiments were determined based on published data on dipteran antennal LDV measurements[8,87,88]. Within-group variation estimates were calculated as part of standard statistical tests and were reasonable for the type of recordings.

Statistical tests for normality (Shapiro-Wilk test with a significance level of $p < 0.05$) were used for each LDV dataset. These were generally found to be non-normally distributed; thus, median and median absolute deviation values are reported throughout.

For the fibrillae erection experiments and amitraz exposure experiments, the Chi-squared test was used to compare the frequencies of fibrillae erection and mechanical states across the different compound injections and circadian time points categories.

For the free fluctuations data, force-step and frequency-modulated sweep stimulation, Wilcoxon signed-rank test was used to perform pairwise comparisons between different compound injections, circadian time points and genotypes, in female and male mosquitoes. The Holms procedure was used to correct for multiple comparisons.

All statistical tests were done in R 4.2.2.

## Reporting summary

Further information on research design is available in the Nature Portfolio Reporting Summary linked to this article.

## Data availability

The RNA-Seq raw data generated in this study have been deposited in the Gene Expression Omnibus (GEO) database under accession code GSE235286. The processed RNA-Seq data and raw data underlying graphs are provided in the Supplementary Information/Source Data file. All data used for analyses in this paper, as well as further details regarding experimental or analytical procedures, are available from the authors. Source data are provided with this paper.

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

## Acknowledgements

We would like to thank you, Shahida Begum, London School Hygiene and Tropical Medicine and Andrea Crisanti from Imperial College London, for providing us with mosquito eggs at the beginning of the study. We would also like to thank you Roberto Galizi for the support on mutant generation design. This work received funding from UK Research and Innovation under the Future Leaders Fellowship scheme (Grant reference MR/S015493/1 to M.A.), from European Union's Horizon 2020 research and innovation programme (Fellowship 752472 to M.A.), from European Research Council (ERC) under the Horizon 2020 research and innovation programme (Consolidator Grant agreement No 648709, to J.T.A.), two UCL Global Challenges Research Fund (GCRF) small grants (to J.T.A. and M.A., respectively), a Wellcome ISSF grant (204841/Z/16/Z) to M.A., a grant from the Biotechnology and Biological Sciences Research Council, UK (BBSRC, BB/V007866/1 to J.T.A.) and a grant from The Human Frontier Science Program (HFSP grant RGP0033/2021 to J.T.A.). This work was also supported by the ANTI-VeC Pump-Priming Grant AV/PP0028/1 to J.T.A. (related to UKRI Grant: BB/R005338/1 and GNCA Grant: EP/X527749/1). This publication was also supported by the project Research Infrastructures for the control of vector-borne diseases (Infravec2), which has received funding from the European Union's Horizon 2020 research and innovation program under grant agreement No 731060. Embryo microinjections and molecular characterisation of transformants were performed by Crisanti Lab at Imperial College London, for which we thank Prof Andrea Crisanti, Dr Roya Haghighat-Khah, Dr Carla Siniscalchi, Dr Rocco D'Amato, Louise Marston and Bathsheba L Gardner.

## Author contributions

M.G., J.S., M.S., and M.A. performed RNA-Seq experiments. M.G. analysed the RNA-Seq data. M.A. generated the transgenic mosquito lines. D.E., D.T.D. and M.A. conducted auditory tests. A.A., J.A. and M.A. analysed auditory test data. J.B. and S.T. maintained the mosquito lines. W.N. and S.M. conducted amitraz experiments and contributed to the experimental design. J.A. and M.A. designed and supervised the project.

## Competing interests

The authors declare no competing interests.
