## [Peer Review File · Nature Communications]

REVIEWER COMMENTS

Reviewer #1 (Remarks to the Author):

The manuscript entitled "A beta-adrenergic-like octopamine receptor modulates the audition of malaria mosquitoes and serves as insecticide target" seeks to establish a role for octopamine signaling in modulating auditory mechanics and sensitivity in *Anopheles gambiae*. The approach is sound, the results internally consistent and findings support the conclusions. However, there were many points in which insufficient details were provided and overall, a model would help the reader frame the many results into a coherent framework. Of greatest concern is the lack of information as to which statistical tests were performed at each point in the manuscript and the actual results from those tests throughout the manuscript. I do not believe that new experiments are needed to further the goal of this study, so in theory my concerns can all be addressed by revising the text to provide greater detail.

Major Concerns

- The statistical tests used are described in the methods but not within the text for each comparison. The figures lack any indication of statistical significance and other than a few chi-square tests, the results of the statistical tests are not provided within the text or the figures. The complete details of every statistical test need to be provided for every comparison and statistical differences need to be indicated in each figure. In other words, it is not sufficient to merely say that something increased or decreased without providing any details of the statistical assessment.
- The box and whisker plots are frequently organized so that the groups directly compared are not next to each other. For instance, the authors state that 1mM octopamine injection increases SSO frequency at ZT4 relative to sham injected and no injection controls (Fig. 3C). If this is the statistical comparison being made, these datasets should be next to each, however in figure 3c each drug treatment has the ZTs next to each other for comparison. The groups to be statistically compared should be next to each other in each box and whisker plot.
- Too little detail is provided in the results section describing figure 2. This would be a good opportunity for the authors to describe the significance of each of their measures.
- A final model depicting the transitions between the different states and the hypothesized contribution of each octopamine receptor would help the reader synthesize a conceptual framework for the entire study. The results as described are relatively internally consistent which should lend well to a schematic, but there are some nuances (such as different contributions from each receptor) that would be easier to follow if they were all depicted together.
- Overall, the discussion was more of a synopsis of the results with very little framing of the findings within the context of the literature. For instance, what have the outcomes been for using amitraz in the field? What about other octopamine signaling targets? What are the implications for modulation of the biomechanics of audition relative to modulation of auditory processing? There are a lot of opportunities to discuss this work within the context of other studies that would help to broaden the general interest in this study for people outside of mosquito biology.
- How does 1mM and 10mM octopamine compare to circulating levels described in different species?
- To make life easier for the field of insect GPCRs, the authors should follow the convention of using established acronyms for orthologous GPCRs and putting the "Ag" (for *Anopheles gambiae*) to designate the species. For instance, "AgOAMB" makes it much easier for a reader to know that this is the *Anopheles gambiae* orthologue of OAMB, than "AGAP000045". This then places the present study within the context of all of the studies on this particular GPCR.
- There needs to be greater discussion of the role of the SSO and quiescent states in amplifying female flight tones. Although the authors state that their findings do not contradict prior suggestions that SSO amplify female flight tones, but their explanation needs more detail. The results from this study strongly suggest that the SSO state does not occur during swarming, so more clarification is required to reconcile these difference conclusions.
- Figure 3A includes the % baseline fibrillar states for each ZT, but Fig 3B does not include baseline flagellar mechanical states. Is it possible to add this to figure 3B?

Minor Concerns:

- Line 53. "A putative efferent neuromodulator of with such function is the octopamine." Perhaps it should read "A putative efferent neuromodulator that serves such a function is octopamine"

Although it would be best to clarify the precise function.

-Line 56. Missing period after "...induced by octopamine"

-In figure 1e, what are the two cycling genes from females (AGAP002560 and AGAP00803)? Only the identity of the male gene is stated.

-Line 293. In addition to 16,000 neurons there are also support cells that could be impacted by neuromodulators which could impact neuronal function.

-Line 298. The word "innervate" can't be used here. The authors have shown that octopamine and serotonin receptors are expressed by cells in the second antennal segment, but they would need to either perform immunocytochemical labeling to demonstrate that it is "innervated" by octopamine or serotonin releasing neurons in this species. They can say that there are cells that are potentially receptive to these signaling molecules but innervate means that neurons releasing these compounds project into the second antennal segment, which wasn't assessed here. Please note that I am not suggesting this be done, rather than the sentence be rephrased.

-Line 322. What is meant by "overflow" here? Do the authors mean that the concentrations of octopamine used may be higher than those found endogenously?

-Figure 1C, D. "OlfactR" should be moved below each trace to avoid occluding the text.

-Figure 2 legend has far too little detail. Each figure should have sufficient detail provided such that it can stand on its own.

-The figure 6 legend lacks a description of panel "e".

Reviewer #2 (Remarks to the Author):

Nature communications review

This is a very comprehensive study on insect hearing and the authors are congratulated on the effort.

This is high quality research on the main vector of malaria.

Some points for improvement:

Some figures in yellow are unreadable and there are mistakes in at least a Fig legend with respect to referring to the proper panels, see details below. I do not know why the authors did not put a star for statistical significance. The P values are not placed in the figure legends either.

The number of mosquitoes used for recordings is not mentioned upfront. They indicate this in the methods but maybe in the first results figure it should be mentioned in the legend so the reader knows all were done with 5 mosquitoes.

I am not sure about the journal style but the subtitles are all in present tense. My understanding is that findings should be in past tense in the first publication because these are not known facts yet. The supplementary tables in Excel cvs, require explanation, see below. The reader should not assume anything, you must explain with a legend, title, or footnote to make the data more accessible on what is what.

Only one of the octopamine receptors showed cyclic transcript expression AGAP000045 while the other receptor that appears to have more influence on hearing at swarm time was not detected as having cyclic transcript expression during the day. What about those signaling systems that had lowest expression at swarm time? Please comment on that and you can fairly say they will be the topic of subsequent research, but you are leaving us with the impression that significant regulatory networks only must peak at swarm time.

Abstract

Add "mating" before partner in line 32.

Introduction

Line 44: Across insects, a remarkable and unique feature of the Johnston's organ, the mosquito ear, is its efferent innervation (do not use parenthesis)

Line 45 it is confusing as written. Write "Input from efferent neurons from the brain seems to be...(for the non-expert)

Line 56. Add period after octopamine 10. Line57 delete "has been shown to" and directly write "It conveys circadian.

Line 59. Not true that Octopamine is restricted to invertebrates. Para-Octopamine has indeed been

measured in humans although is a poor ligand of adrenergic receptors and has been measured in other mammals. One way out is to say octopamine is the most abundant biogenic amine in invertebrate systems or something like that.

Line 77 composed of two main elements, a sound receiver, the antennal flagellum, and ...

Because this journal is not for entomologists, otherwise is not clear if the sound receiver is a different element than the antennal flagellum.

Line 82: delete parenthesis of "hosting the JO" (add the) and in line 85 delete comma after male and delete "i.e. transcripts" as the sentence reads better without that.

The supplementary tables 1 and 2 do not have any designation as to being from male or female despite that in line 87 under Results it says so, but one is not sure which one is which. If 1 is male and 2 is female, then they need to add "respectively".

Line 97: , and had significantly higher expression in male than female. (delete: more expression)

Line 122: the most interesting part is mentioned in Supplementary Tables 6 and 7. When I opened these tables they are unclear as one does not know what BH.Q and ADJ.P, Per, LAG, AMP means, these are the different columns. Not everyone interested in this should be an expert. Otherwise indicate where those terms are described in the manuscript.

The authors need to put a legend, title, or footnote in this table before the table itself explaining what all of this means.

In Fig. 1 e the authors should add below the AGAP...ID the actual annotation like Orco GluR, and the OAMB receptor. In Panel C the first yellow cannot be read at all, I printed the figure and it is unreadable. Must change color, even to black. The second yellow is difficult to read but I can see PeptideR. For the first I cannot read anything. The authors must mention the categories under c and d.

In Table 1 because there are so many genes the authors should bold or mark somehow the main sequences discussed the Oamb, the Ir93a, the GluRIIA the Orco AGAP002560. It is frustrating to have to look at the whole list to find those few.

Also why is the DopEcR under Photoreceptors (Dopamine/Ecdysone)? I am not familiar with this, please cite.

Fig. 2 panel C in the orange box I think stiffness is misspelled. You may want to increase the font at least 1 number. I think it says stffiness. The figure legend should give some interpretation of the results, make clear panels C and D are examples of the type of recordings. Panel D needs some basic short description for the non-expert.

Line 136 Add (Fig. 3) at the end of the line aftergambiae (Fig. 3).

In the legend of Fig 3c define what is n number of what? If the differences between ZT4 and ZT12 are statistically significant why not put a star between the red and green boxes. The small numbers in the gray boxes in red and green should be 1 or 2 size font bigger.

In Fig 3 the colors chosen to differentiate partially erected and fully erected are very similar, please choose a much lighter blue for the collapsed and an intermediate color for the partially erected. Even when printed they look the same. The green shown for mechanical state is better.

Fig 4 the numbers in the small gray boxes are illegible. You have space make them bigger. Orange was a poor choice of color for numbers and graph, can you use violet or magenta?

L197 add (Fig. 4a,b).

Fig. 4 legend it says Red boxplots show ZT4 data and green ZT12 but I see blue and orange (?). Maybe they got confused with another previous figure or version? Fig 3c is in Red and green.

Line 923. ...that was acuter. Is this is a common usage? Or (suggestion) "that was more pronounced during swarm time.."

Line 206. For consistency change to

"Octopamine injection had milder effects on female Anopheles..

Line 212: biophysical parameters or biophysical variables? What you measure is a variable not a parameter. Or Biophysical parameter values? You have this later in the manuscript.

In Fig 6 the numbers in yellow are totally unreadable change the yellow color for AGAP000045- to

something else. These numbers are important because we do not see statistical significance or not in the graphs.

The legend of Fig. 6 is wrong in that there should be a b) in line 943 as follows: b) A cassette...Then all the following letters should be change accordingly so the current b) should be c) Fibrillae...d) Flagellar...e) SSO frequencies...

If known, please indicate at what time of the day were these mutants tested with octopamine in Fig 6e because even if one gene is mutated the other receptor is still there and could influence the results. I am not sure if the ZT12 still applies for the SSO frequency and SSO amplitude. Indicate on the graphs or in the figure legend.

Manuscript Line 241: it should say Fig. 6e not Fig 5d.

In figure legend of Fig 7 lines 973-974 need to be re-written because it says that "the increase in the stiffness values was very faint in octopamine receptor mutants", but indeed there was only a minor increase in the -45 mutant while the -2886 decreased. Re-write as this appears to contradict what the figure shows. Also faint (?) maybe choose another word.

In Fig 8b it would help the reader if the legend for dark blue erected were on the left below the blue bar and the collapsed light blue below the mutant, to emphasize that the wt has erected fibrillae (suggestion).

Discussion

L300 Delete "has been shown to" and just write "Serotonin modulates mating behaviour. Circadianicity not only can be expressed by genes that PEAK in higher expression but also by those that lower in expression as roles can be either inhibitory or promoting gene networks. Any comments on this?"

Do not use parenthesis that much, it is not needed, especially when clarifications are necessary. So, delete parenthesis in line 320 and just put a comma. Or write i.e. reduction of...This is important because the following sentence says "This effect...which refers to the effect in parenthesis!"

Line 324 should you write "reflecting the higher octopamine receptor(s) expression? You implicated two receptors.

Throughout the paper the authors use the term "octopamine exposure"; in the introduction they start indicating efferent innervation from the brain with the role of octopamine as a neuromodulator or neurotransmitter. The experiments done here are by body injection of octopamine in which octopamine is mimicking the effect of a neurohormone, and this is touched upon in lines 340-341 of the discussion. I suggested in the results section to use octopamine injections but maybe in the discussion the authors may want to comment on the role as neurohormone analyzed in this study. Eventually genetic technologies could evolve to do optogenetics in mosquito where the role of neurotransmitters or neuromodulators could be more "cleanly" studied.

Although the gene KO of receptors strongly supports their claims, octopamine could have undoubtedly affect the release of other peptides or neurotransmitters when broadly injected. This is maybe seen in the half-effects of the 000045 KO phenotype?

The role of neurohormones in females is discussed in line 360.

Please re-write lines 362-365. It seems grammatically wrong and it is not really clear what you are trying to say. Maybe "Transcriptomic and structural analyses support that the female ear is highly complex but new approaches are needed to elucidate the role of female hearing during mating, as the approaches utilized to study male hearing have not yielded new insights. Developing..." something like that because the "we are missing something" is too colloquial to me.

At the end of the discussion, you mention "insecticides". First it is not clear how the dose you use for your experiment (concentration) compares to the concentration of amitraz when applied as insecticide/acaricide. You also need to find a citation where amitraz has been tested in mosquitoes to indicate what the LC50 or LD50 is. In the bioassays indeed "exposure" was for a short time. Here exposure is correctly used but do not use the word "exposure" when you are referring to the injected octopamine, it could result in confusion.

Second, as amitraz is an agonist I do not know if having the stiff fibrillae will promote or

discourage mating. It seems to me that you want to reproduce the KO phenotype for what you will need an antagonist of the octopamine receptor, so if this is the case you must say it. Therefore, it may not be necessarily an insecticide but a "mating disruptor" and not necessarily a killer of mosquitoes, but more like the ones that have been applied to disrupt mating in agricultural pests with pheromones. So, lines 388-391 are confusing or contradictory because if they are insecticides we do not care if disrupt reproduction because they will kill. You talk about resistance; one way to avoid resistance is to have two different modes of action at distinct target sites, so the probability of target site insensitivity is decreased. It is also a question of dose. Or just a cocktail of an insecticide with a mating disruptor? These hearing disruptors could be applied at lower dose with an insecticide of different mode of action? Re-write the paragraph 388-391 for clarity. The problem is line 388. And maybe instead of targets for insecticide development you must say "Novel targets for novel interventions is of highest priority..."

Methods: can you please add the LIGHTS/lamps used in the Percival cabinets, what type of tubes were used? Intensity, anything that informs the light regime.
L410 Eppendorfs with capital E.
It is not clear who prepared the Library (line 415). Was it done in house by authors or sent to some facility? Indicate.

I cannot comment on the transcriptomic analyses or the formulas in pgs 13-14.
Sigma sells different p-octopamine so please add the catalog number and indicate if it was the analytical standard or what. Also, for clarity write para-octopamine hydrochloride (Sigma-Aldrich catalog #).
Similarly, for Amitraz I suggest writing the correct name or at least say if it was the Analytical standard and the NSC number below.
N'-(2,4-Dimethylphenyl)-N-[[2,4-dimethylphenyl]imino]methyl}-N-methylmethanimidamide, NSC 324552

Compound injection: was this done manually or with an injector apparatus?
I do not think the injection procedure is reproducible as described. If needed add a photograph as suppl information. I could not reproduce your description. What volume you injected approx.?

References

Check for consistency some have many authors some just say et al E.g number 52.
Ref 54 Belton has misspellings in two words.
Not sure about journal rules, but all scientific names of insects are not in italics and normally they should be.

RESPONSE TO REVIEWERS' COMMENTS

Reviewer #1 (Remarks to the Author):

The manuscript entitled “A beta-adrenergic-like octopamine receptor modulates the audition of malaria mosquitoes and serves as insecticide target” seeks to establish a role for octopamine signaling in modulating auditory mechanics and sensitivity in *Anopheles gambiae*. The approach is sound, the results internally consistent and findings support the conclusions. However, there were many points in which insufficient details were provided and overall, a model would help the reader frame the many results into a coherent framework. Of greatest concern is the lack of information as to which statistical tests were performed at each point in the manuscript and the actual results from those tests throughout the manuscript. I do not believe that new experiments are needed to further the goal of this study, so in theory my concerns can all be addressed by revising the text to provide greater detail.

We thank the reviewer for the support of the work and for the very useful comments that have helped us to improve the manuscript. We have implemented all comments suggested, including i) providing a clearer interpretation of octopamine’s auditory role, by suggesting models that facilitate the understanding of the results and framing them into what is known about the acoustic detection of mosquito mating partners, ii) including information on statistical tests in the manuscript and figures, iii) providing greater detail of some aspects that have been raised. A detailed response to individual comments is provided below:

Major Concerns

-The statistical tests used are described in the methods but not within the text for each comparison. The figures lack any indication of statistical significance and other than a few chi-square tests, the results of the statistical tests are not provided within the text or the figures. The complete details of every statistical test need to be provided for every comparison and statistical differences need to be indicated in each figure. In other words, it is not sufficient to merely say that something increased or decreased without providing any details of the statistical assessment.

Thanks for this suggestion. The results of all statistical test were provided in the supplementary materials. We have now included the results of statistical tests in the figures and in the main text.

-The box and whisker plots are frequently organized so that the groups directly compared are not next to each other. For instance, the authors state that 1mM octopamine injection increases SSO frequency at ZT4 relative to sham injected and no injection controls (Fig. 3C). If this is the statistical comparison being made, these datasets should be next to each, however in figure 3c each drug treatment has the ZTs next to each other for comparison. The groups to be statistically compared should be next to each other in each box and whisker plot.

We have rearranged Figure 3 to place the groups to be compared next to each other.

-Too little detail is provided in the results section describing figure 2. This would be a good opportunity for the authors to describe the significance of each of their measures.

We have added a paragraph describing the significance of the auditory tests, we have also expanded the legend in Figure 2 to facilitate the understanding of the auditory tests conducted:

Line 134: *“In a nutshell, individual tests quantified different functionally relevant mechanical properties of the mosquito ear. It should be noted that all measured represent compound responses, i.e. all mechanical parameters (e.g. flagellar best frequency or tuning sharpness) reflect the sum total*

of all components mechanically connected to the flagellum. Taken together these tests will enable better reconstructions, or predictions, of mosquito auditory sensitivity in different behavioural or ecological contexts. At present it is e.g. not known in which contexts male flagellar ears enter the SSO state, respectively. Likewise, the sensory-ecological control of fibrillae erection is still not fully clear. Yet the state of all these parameters will have crucial effects on the sensitivity and tuning of the mosquito ear. We thus assessed the pattern of fibrillae erection and quantified the properties of both stimulated and unstimulated ears. We used frequency sweeps to determine the auditory properties in response to more naturalistic stimuli and we used force step actuation as more analytic stimuli to extract principal parameters of flagellar mechanics (e.g. flagellar stiffness)."

-A final model depicting the transitions between the different states and the hypothesized contribution of each octopamine receptor would help the reader synthesize a conceptual framework for the entire study. The results as described are relatively internally consistent which should lend well to a schematic, but there are some nuances (such as different contributions from each receptor) that would be easier to follow if they were all depicted together.

We agree with the reviewer that a model for the function of the receptors would help the reader. Our results show that AgOct β 2 is the primary receptor mediating octopamine auditory role, as octopamine-induced responses are abolished in knockout mosquitoes. AgOAMB seems to modulate finer responses related to the mechanical auditory tuning, as AgOAMB knockouts show a phenotype in the SSO frequency. Both receptors use different second messengers (AgOct β causes an increase of cAMP and AgOAMB an increase of Ca²⁺), increasing potential physiological effects. Due to the complexity of octopamine release in the ear, it can also be that both receptors are located in different regions of the neurons, or even in different cells (different neuronal type in the JO, support cells). We have modified the discussion to provide a model of how both receptors might be acting complementary to modulate the auditory function of the male during swarm time. To develop a more precise model, we would need further experiments to test for example changes in sensitivity levels of the ear, or to conduct patch clamp experiments of single auditory neurons that would allow to test for the function of both receptors at single-neuron level. The text now reads (lines 411-447):

"In this study, we identified the receptors mediating octopamine's auditory role. AgOct β 2 is a β -adrenergic-like octopamine receptor orthologue of Octb2R in other insects 66. Most octopamine-induced auditory responses observed, including the erection of the antennal fibrillae and the flagellar mechanical effects, were nearly abolished in AgOct β 2- mutants, supporting its role as the primary octopamine receptor in the mosquito ear (Fig. 6, 7). The other octopamine receptor detected, the α -adrenergic-like octopamine receptor AgOAMB, showed a peak of expression at swarm time, which probably accounts for the stronger effects of octopamine at swarm time for most parameters studied. AgOAMB displayed an intermediate phenotype, with octopamine-mediated auditory effects being reduced compared to wildtype animals, both regarding the fibrillae erection and the flagellar stiffness. Interestingly, none of the receptor mutants ceased SSOs upon octopamine injections (Fig. 6d), supporting our previous argument that, in nature, octopamine would modulate SSOs rather than stop them if, for example, not both receptors are activated at the same time by the octopaminergic efferent innervation.

Also, the auditory responses of AgOAMB mutants can be misleading and need to be discussed. The fact that some biophysical parameters extracted from frequency-modulated sweep stimulation show stronger value shifts upon octopamine injection in AgOAMB mutants compared to wildtypes (e.g. peak and oscillator frequency, Fig. 7) does not mean that octopamine has a stronger effect on AgOAMB mutants. Instead, octopamine effect is stronger in wildtypes and therefore it pushes flagella to quiescent states, so they are removed from the dataset in the graph (Fig. 6, 7) that shows only SSO responses. In AgOAMB mutants, because the stiffness shift is weaker, mosquitoes can maintain SSOs, and this is reflected in stronger shifts in the other biophysical parameters that are nonetheless not

powerful enough to induce SSO cessation, as it occurs in wildtypes. In AgOct β 2 mutant mosquitoes, SSO values barely changed after octopamine injection. Moreover, our data support that the function of these receptors is non redundant and complementary, as knocking out either receptor shows distinct auditory phenotypes. They also support some constitutively signalling of AgOct β 2 receptor during swarm time, as baseline differences in some auditory parameters are present when compared to wildtypes. It is plausible that AgOct β 2 plays a more fundamental role preparing the male ear for swarm time, while AgOAMB modulate finer adjustment of the mechanical tuning to adapt the ear to the distinct sensory ecology of the specific swarm where the male mosquito is trying to find a female.

The fact that β -adrenergic-like (AgOct β 2) and α -adrenergic-like (AgOAMB) octopamine receptors use different second messengers, namely an increase in Ca²⁺ and an increase in cAMP 12, respectively, further supports a functional distinct role of both octopamine receptors. These two octopamine receptors are also found in mechanosensory neurons of the spider *Cupiennius salei* that are innervated by efferent octopaminergic fibres 67. In the spider mechanosensory neurons, each receptor modulates different aspects of neuronal sensitivity, including baseline levels and rapid changes 68. Further experiments are necessary to disentangle the specific auditory role of each receptor in mosquitoes, and factors to consider include whether they localize to different neuronal regions, to different cells (e.g. different neurons in the scolopidia or support cells) or if they are involved in temporal changes in the auditory physiology. “

-Overall, the discussion was more of a synopsis of the results with very little framing of the findings within the context of the literature. For instance, what have the outcomes been for using amitraz in the field? What about other octopamine signaling targets? What are the implications for modulation of the biomechanics of audition relative to modulation of auditory processing? There are a lot of opportunities to discuss this work within the context of other studies that would help to broaden the general interest in this study for people outside of mosquito biology.

We thank you the reviewer for this comment. We agree that there are many points that would benefit from a wider discussion, and we have tried to expand the discussion regarding some of the points mentioned by the reviewer, especially concerning amitraz implications for mosquito control (lines 448-464), octopamine receptor signalling (lines 411-447) or about the implications of modulation of the biomechanics in mosquito mating partner detection (lines 329-367). We appreciate that the discussion of our results can be approached from different angles. Due to constraints in space, we have tried to include those that we believe are more relevant and interesting for the research community, but we are aware that we might have left some significant topics out. We apologize for this and we are open to further suggestions.

-How does 1mM and 10mM octopamine compare to circulating levels described in different species?

Previous literature in larvae of the moth *Mythimna separata* show octopamine circulating levels of ~200pg/ml (Kong et al. 2018). Another paper reports levels of ~400pg/ml in the haemolymph of the field cricket *Gryllus bimaculatus* (Adamo, Linn, and Hoy 1995). Our concentrations were higher than this, as 1mM octopamine hydrochloride corresponds to 0.189 mg/ml, but the concentrations we have used are comparable to other studies (Fuchs et al. 2014; Fussnecker, Smith, and Mustard 2006; Barron et al. 2007; Stevenson et al. 2005). It is also important to mention that previous research shows that octopamine does not reach the mosquito JO through the haemolymph, but rather via efferent fibers that innervate the auditory cilia region and target the receptor via volume transmission (Andrés et al. 2016; Loh et al. 2023). The octopamine concentration in the extracellular space after octopamine secretion would be high.

-To make life easier for the field of insect GPCRs, the authors should follow the convention of using established acronyms for orthologous GPCRs and putting the “Ag” (for Anopheles gambiae) to designate the species. For instance, “AgOAMB” makes it much easier for a reader to know that this is

the *Anopheles gambiae* orthologue of OAMB, than “AGAP000045”. This then places the present study within the context of all of the studies on this particular GPCR.

This has now been corrected. AGAP002886 has been named AgOct β 2 and AGAP000045 AgOAMB.

-There needs to be greater discussion of the role of the SSO and quiescent states in amplifying female flight tones. Although the authors state that their findings do not contradict prior suggestions that SSO amplify female flight tones, but their explanation needs more detail. The results from this study strongly suggest that the SSO state does not occur during swarming, so more clarification is required to reconcile these difference conclusions.

Thanks for this suggestion. We agree that this is a key point to interpret our findings and to suggest a role for octopamine in modulating the mosquito auditory function in the swarm. We have added some more points to the discussion around SSOs and female detection (lines 336-362):

*“We also detected an effect of octopamine in modulating the flagellar mechanical state by promoting quiescent states over SSOs (Fig. 3b). SSOs have been linked to the amplification of female flight tones in the swarm 5,6,8 It seems therefore that octopamine acts on two competing frontiers: one that modulates the SSO to higher frequencies which favours female audibility 58 while at the same time promoting the opposing mechanical state of the flagellum into quiescence. We could infer that there is a fine balance based on the total concentration of octopamine within the system. However, we should first consider that our knowledge of SSO mechanism and function is still incomplete. The SSOs seen in the flagellar ears of male *Anopheles* mosquitoes are unique across auditory systems, both in magnitude and properties 8. The fact that their energy content varies by several orders of magnitude across different mosquito species 8 also points to substantial degrees of ecological diversification. It seems beyond doubt that SSOs are a crucial component of the mosquito hearing mechanism, but it will require further studies dedicated to SSOs to explore them on all relevant functional levels (e.g. within the entire plane of flagellar mobility and binaurally, across both ears) and to finally understand how they support the male ear in the detection of faint female flight tones. Here, circadian modulations mediated by octopamine through different, functionally distinct, receptors clearly play a vital role but there are still many unknowns. We would argue that in our experimental setup, octopamine effects on the mechanical state are too severe compared to its natural role in the auditory system, where it probably just modulates certain SSO parameters (e.g. frequency and amplitude). In our experiments, injected octopamine reaches the mosquito ear through the haemolymph rather than via efferent terminals, activating all octopamine receptors at once. Therefore, the spatial and time resolution of the system is altered, potentially inducing the cessation of SSOs rather than a modulation of their properties. Moreover, gluing the mosquito ear for our auditory tests might induce stress responses that can affect the flagellum, and SSOs (e.g 59). Our results are consistent with the canonical view that the active processes found to support hearing across taxa (such as e.g. SSOs in mosquitoes) only operate within a certain space of biophysical values and show critical dependence on the state of distinct, yet mostly unidentified, control parameters; octopamine may very well be one of those. We suggest that although injected octopamine induce SSO cessation this is not necessarily what occurs in nature.”*

-Figure 3A includes the % baseline fibrillar states for each ZT, but Fig 3B does not include baseline flagellar mechanical states. Is it possible to add this to figure 3B?

We have added the flagellar mechanical states at the baseline, and modify the manuscript and the legend text accordingly.

Minor Concerns:

-Line 53. "A putative efferent neuromodulator of with such function is the octopamine." Perhaps it should read "A putative efferent neuromodulator that serves such a function is octopamine" Although it would be best to clarify the precise function.

This has been corrected:

Line 56: "A putative efferent neuromodulator that serves an auditory function is octopamine."

-Line 56. Missing period after "...induced by octopamine"

This has been corrected.

-In figure 1e, what are the two cycling genes from females (AGAP002560 and AGAP00803)? Only the identity of the male gene is stated.

We have added this information in Fig. 1 legend.

Line 975: "A single gene in males, the α -adrenergic-like octopamine receptor AGAP000045, and two genes in females, the odorant receptor co-receptor *Orco* (AGAP002560) and ionotropic glutamate receptor subunit GluRIIIa (AGAP000803) show cyclic expression in the mosquito ear."

-Line 293. In addition to 16,000 neurons there are also support cells that could be impacted by neuromodulators which could impact neuronal function.

We agree with the reviewer. We have included a sentence (line 318): "It is also plausible that the neuromodulators target support cells in the JO to affect the mechanotransduction process (Andrés et al. 2014; Prelic et al. 2021; Li Zheng, Adams, and Chisholm 2020)"

-Line 298. The word "innervate" can't be used here. The authors have shown that octopamine and serotonin receptors are expressed by cells in the second antennal segment, but they would need to either perform immunocytochemical labeling to demonstrate that it is "innervated" by octopamine or serotonin releasing neurons in this species. They can say that there are cells that are potentially receptive to these signaling molecules but innervate means that neurons releasing these compounds project into the second antennal segment, which wasn't assessed here. Please note that I am not suggesting this be done, rather than the sentence be rephrased.

This makes sense. We have changed "show" for "suggest". The sentence reads (line 321): "*Our transcriptomic analyses suggest that, as previously described in Cx. quinquefasciatus mosquitoes (Andrés et al. 2016), the biogenic amines octopamine and serotonin also innervate the malaria mosquito ear.*" We think using "suggest" is justified as it has previously been shown that octopamine and serotonin innervate the JO in *Cx. quinquefasciatus* mosquitoes (Andrés et al. 2016) and that the pattern of efferent innervation is conserved across mosquito species (Su et al. 2018).

-Line 322. What is meant by "overflow" here? Do the authors mean that the concentrations of octopamine used may be higher than those found endogenously?

Yes, that was what we meant. We have rewritten this paragraph and it is quite different now, but a sentence containing this information is (lines 354-357): "*injected octopamine reaches the mosquito ear through the haemolymph rather than via efferent terminals, activating all octopamine receptors at once. Therefore, the spatial and time resolution of the system is altered, potentially inducing the cessation of SSOs rather than a modulation of their properties.*"

-Figure 1C, D. "OlfactR" should be moved below each trace to avoid occluding the text.

We have corrected this.

-Figure 2 legend has far too little detail. Each figure should have sufficient detail provided such that it can stand on its own.

A more detailed legend has been added to Figure 2.

-The figure 6 legend lacks a description of panel "e".

This has been added.

Reviewer #2 (Remarks to the Author):

Nature communications review

This is a very comprehensive study on insect hearing and the authors are congratulated on the effort. This is high quality research on the main vector of malaria.

We thank the reviewer for the support and valuable comments that have helped us to greatly improve the manuscript.

Some points for improvement:

Some figures in yellow are unreadable and there are mistakes in at least a Fig legend with respect to referring to the proper panels, see details below. I do not know why the authors did not put a star for statistical significance. The P values are not placed in the figure legends either.

We have included the p values in the figures and changed the colour coding as suggested by the reviewer.

The number of mosquitoes used for recordings is not mentioned upfront. They indicate this in the methods but maybe in the first results figure it should be mentioned in the legend so the reader knows all were done with 5 mosquitoes.

We have included this information in the figure legends. The mosquito numbers were 5-11 depending on the experiments (5 were the mosquitoes included in each vial for the circadian entrainment).

I am not sure about the journal style but the subtitles are all in present tense. My understanding is that findings should be in past tense in the first publication because these are not known facts yet.

We have corrected most subtitles to be written in past tense. We have just left in present tense those that were general conclusions of the results presented.

The supplementary tables in Excel cvs, require explanation, see below. The reader should not assume anything, you must explain with a legend, title, or footnote to make the data more accessible on what is what.

Thanks for this comment, we have included legends for all supplementary tables as well as improved their annotation.

Only one of the octopamine receptors showed cyclic transcript expression AGAP000045 while the other receptor that appears to have more influence on hearing at swarm time was not detected as having cyclic transcript expression during the day. What about those signaling systems that had lowest expression at swarm time? Please comment on that and you can fairly say they will be the topic of subsequent research, but you are leaving us with the impression that significant regulatory networks only must peak at swarm time.

We agree with the reviewer that genes showing lowest expression at swarm time could also play an important regulatory role. Our analysis to detect cyclic expression would identify any gene whose expression is rhythmic, independent on whether the expression shows peaks or troughs along the day. Following this pipeline, from the pool of genes belonging to our GO terms of interest, the only gene that showed cyclic expression in the male JO was AgOAMB, and it happened to show a peak of expression at ZT12 during the laboratory swarm time. We would have been able to equally detect any gene showing a trough at ZT12, but this was not the case in the male JO. Indeed, Figure 1 shows that in females, we detected 2 cycling genes, AGAP00803, which shows a peak of expression at ZT12 and AGAP002560, which shows the lowest expression at ZT12.

Abstract

Add “mating” before partner in line 32.

This has been added.

Introduction

Line 44: Across insects, a remarkable and unique feature of the Johnston’s organ, the mosquito ear, is its efferent innervation (do not use parenthesis)

This has been corrected.

Line 45 it is confusing as written. Write “Input from efferent neurons from the brain seems to be...(for the non-expert)

This has been included.

Line 56. Add period after octopamine 10. Line57 delete “has been shown to” and directly write “It conveys circadian.

This has been corrected.

Line 59. Not true that Octopamine is restricted to invertebrates. Para-Octopamine has indeed been measured in humans although is a poor ligand of adrenergic receptors and has been measured in other mammals. One way out is to say octopamine is the most abundant biogenic amine in invertebrate systems or something like that.

We have corrected the sentence to “octopamine signalling is mostly restricted to invertebrates” (line 62).

Line 77 composed of two main elements, a sound receiver, the antennal flagellum, and ... Because this journal is not for entomologists, otherwise is not clear if the sound receiver is a different element that the antennal flagellum.

This has been added.

Line 82: delete parenthesis of “hosting the JO” (add the) and in line 85 delete comma after male and delete “i.e. transcripts” as the sentence reads better without that.

The sentence has been corrected as (line 86): “We collected exclusively second antennal segments, hosting the JO, without flagella”.

“i.e. transcripts” has been deleted.

The supplementary tables 1 and 2 do not have any designation as to being from male or female despite that in line 87 under Results it says so, but one is not sure which one is which. If 1 is male and 2 is female, then they need to add “respectively”.

This has been added.

Line 97: , and had significantly higher expression in male than female. (delete: more expression)

This has been corrected.

Line 122: the most interesting part is mentioned in Supplementary Tables 6 and 7. When I opened these tables they are unclear as one does not know what BH.Q and ADJ.P, Per, LAG, AMP means, these are the different columns. Not everyone interested in this should be an expert. Otherwise indicate where those terms are described in the manuscript.

The authors need to put a legend, title, or footnote in this table before the table itself explaining what all of this means.

Thanks for this comment. We have improved the supplementary table annotation and included legends to all supplementary tables to make their content more accessible.

In Fig. 1 e the authors should add below the AGAP...ID the actual annotation like Orco GluR, and the OAMB receptor. In Panel C the first yellow cannot be read at all, I printed the figure and it is unreadable. Must change color, even to black. The second yellow is difficult to read but I can see PeptideR. For the first I cannot read anything. The authors must mention the categories under c and d.

Both the receptor categories and the annotations have been added to the figure legend and figure.

In Table 1 because there are so many genes the authors should bold or mark somehow the main sequences discussed the Oamb, the Ir93a, the GluRIIA the Orco AGAP002560. It is frustrating to have to look at the whole list to find those few.

These genes have been highlighted in the table.

Also why is the DopEcR under Photoreceptors (Dopamine/Ecdysone)? I am not familiar with this, please cite.

This gene was assigned to this category according to the orthology analysis results, but we agree it would be better described as a dopamine receptor. We have now added it in a different category in the table and Fig 1b,c.

put groups to be compared next to each other

We have changed the arrangement of figures to locate groups to compare next to each other.

panel C in the orange box I think stiffness is misspelled. You may want to increase the font at least 1 number. I think it says stffiness. The figure legend should give some interpretation of the results, make clear panels C and D are examples of the type of recordings. Panel D needs some basic short description for the non-expert.

We have corrected all these points in Fig. 2 following the reviewer suggestions.

Line 136 Add (Fig. 3) at the end of the line aftergambiae (Fig. 3).

This has been added.

In the legend of Fig 3c define what is n number of what? If the differences between ZT4 and ZT12 are statistically significant why not put a star between the red and green boxes. The small numbers in the gray boxes in red and green should be 1 or 2 size font bigger.

We have increased the numbers in the gray boxes, and included the results of the statistical tests In the plot. We have also define n numbers in all figure legends: *“n numbers represent the total number of runs (individual stimulated or unstimulated sections) that passed the curation process and are included in the data analysis.”*

In Fig 3 the colors chosen to differentiate partially erected and fully erected are very similar, please choose a much lighter blue for the collapsed and an intermediate color for the partially erected. Even when printed they look the same. The green shown for mechanical state is better.

We have changed the color pattern in Fig. 3a so that the collapsed fibrillae group are shown in a lighter blue.

Fig 4 the numbers in the small gray boxes are illegible. You have space make them bigger. Orange was a poor choice of color for numbers and graph, can you use violet or magenta?

We have increased the numbers in the gray boxes. We have changed the colors to blue and red to fit the color choice in Figure 3 and make it easier to be distinguished for the reader.

L197 add (Fig. 4a,b).

Added.

Fig. 4 legend it says Red boxplots show ZT4 data and green ZT12 but I see blue and orange (?). Maybe they got confused with another previous figure or version? Fig 3c is in Red and green.

This has been now corrected.

Line 923. ...that was acuter. Is this is a common usage? Or (suggestion) “that was more pronounced during swarm time..”

We have changed the test as suggested.

Line 206. For consistency change to
“Octopamine injection had milder effects on female Anopheles..

This has been changed.

Line 212: biophysical parameters or biophysical variables? What you measure is a variable not a parameter. Or Biophysical parameter values? You have this later in the manuscript.

We have corrected this to “biophysical parameter values”.

In Fig 6 the numbers in yellow are totally unreadable change the yellow color for AGAP000045- to something else. These numbers are important because we do not see statistical significance or not in the graphs.

The color for the mutant AGAP000045- has been changed to dark green.

The legend of Fig. 6 is wrong in that there should be a b) in line 943 as follows: b) A cassette...Then all the following letters should be change accordingly so the current b) should be c) Fibrillae...d) Flagellar...e) SSO frequencies...

The figure legend has been corrected.

If known, please indicate at what time of the day were these mutants tested with octopamine in Fig 6e because even if one gene is mutated the other receptor is still there and could influence the results. I am not sure if the ZT12 still applies for the SSO frequency and SSO amplitude. Indicate on the graphs or in the figure legend.

ZT12 applies for all data on Fig. 6. We have now added (ZT12) close to every graph title to clarify this.

Manuscript Line 241: it should say Fig. 6e not Fig 5d.

This has been corrected.

In figure legend of Fig 7 lines 973-974 need to be re-written because it says that “the increase in the stiffness values was very faint in octopamine receptor mutants”, but indeed there was only a minor increase in the -45 mutant while the -2886 decreased. Re-write as this appears to contradict what the figure shows. Also faint (?) maybe choose another word.

This has been re-written (lines (1142-1143): “ *The sharp increase in the stiffness values observed in wildtype mosquitoes was absent in octopamine receptor mutants. In AgOct82 knock-outs the stiffness values even decreased upon octopamine injections.* ”.

In Fig 8b it would help the reader if the legend for dark blue erected were on the left below the blue bar and the collapsed light blue below the mutant, to emphasize that the wt has erected fibrillae (suggestion).

Thanks for this suggestion. We have implemented this change.

Discussion

L300 Delete “has been shown to” and just write “Serotonin modulates mating behaviour.

This has been corrected.

Circadianicity not only can be expressed by genes that PEAK in higher expression but also by those that lower in expression as roles can be either inhibitory or promoting gene networks. Any comments on this?

We totally agree with this. This relates to the comment that we have added above: Our analysis to detect cyclic expression would identify any gene whose expression is rhythmic, independent on whether the expression shows peaks or troughs along the day. Following this pipeline, from the pool of genes belonging to our GO terms of interest, the only gene that showed cyclic expression in the male JO was AgOAMB, and it happened to show a peak of expression at ZT12 during the laboratory swarm time. We would have been able to equally detect any gene showing a trough at ZT12, but this was not the case in the male JO. Indeed, Figure 1 shows that in females, we detected 2 cycling genes, AGAP00803, which shows a peak of expression at ZT12 and AGAP002560, which shows the lowest expression at ZT12.

Do not use parenthesis that much, it is not needed, especially when clarifications are necessary. So, delete parenthesis in line 320 and just put a comma. Or write i.e. reduction of...This is important because the following sentence says “This effect...which refers to the effect in parenthesis!

This has been implemented.

Line 324 should you write “reflecting the higher octopamine receptor(s) expression? You implicated two receptors.

This section has been re-written following reviewer recommendations.

Throughout the paper the authors use the term “octopamine exposure”; in the introduction they start indicating efferent innervation from the brain with the role of octopamine as a neuromodulator or neurotransmitter. The experiments done here are by body injection of octopamine in which octopamine is mimicking the effect of a neurohormone, and this is touched upon in lines 340-341 of the discussion. I suggested in the results section to use octopamine injections but maybe in the discussion the authors may want to comment on the role as neurohormone analyzed in this study. Eventually genetic technologies could evolve to do optogenetics in mosquito where the role of neurotransmitters or neuromodulators could be more “cleanly” studied. Although the gene KO of receptors strongly supports their claims, octopamine could have undoubtedly affect the release of other peptides or neurotransmitters when broadly injected. This is maybe seen in the half-effects of the 000045 KO phenotype?

The role of neurohormones in females is discussed in line 360.

This is an interesting point that is however difficult to resolve with the available techniques, as indicated by the reviewer. To point at this limitation in the discussion we have added (line 354): *“In our experiments, injected octopamine reaches the mosquito ear through the haemolymph rather than via efferent terminals, activating all octopamine receptors at once. Therefore, the spatial and time resolution of the system is altered, potentially inducing the cessation of SSOs rather than a modulation of their properties.”*

Please re-write lines 362-365. It seems grammatically wrong and it is not really clear what you are trying to say. Maybe “Transcriptomic and structural analyses support that the female ear is highly complex but new approaches are needed to elucidate the role of female hearing during mating, as the approaches utilized to study male hearing have not yielded new insights. Developing...” something like that because the “we are missing something” is too colloquial to me.

We have changed this sentence to: *“However, both the transcriptomic analysis in this paper and previous anatomical studies (Su et al. 2018) support that the female ear is complex but new approaches are needed to elucidate the role of female hearing during mating, as the approaches utilized to study male hearing have not yielded new insights.”*

At the end of the discussion, you mention “insecticides”. First it is not clear how the dose you use for your experiment (concentration) compares to the concentration of amitraz when applied as insecticide/acaricide. You also need to find a citation where amitraz has been tested in mosquitoes to indicate what the LC50 or LD50 is. In the bioassays indeed “exposure” was for a short time. Here exposure is correctly used but do not use the word “exposure” when you are referring to the injected octopamine, it could result in confusion.

We have corrected the word “exposure” by “injection” when referring to octopamine along the manuscript. We have not found information regarding the toxicity of amitraz in malaria mosquitoes (we are currently investigating this, and we will publish the results soon). The toxicity in larvae of the dengue mosquito has been evaluated (LC50 of 323 µg /ml, (Ahmed and Matsumura 2012)). We based our amitraz concentrations in the FAO guidelines “Guidelines resistance management and integrated parasite control in ruminants” (‘Food and Agriculture Organization of the United Nations, 2004. Guidelines Resistance Management and Integrated Parasite Control in Ruminants’). We have added this information to Fig. 8 legend.

Second, as amitraz is an agonist I do not know if having the stiff fibrillae will promote or discourage mating. It seems to me that you want to reproduce the KO phenotype for what you will need an

antagonist of the octopamine receptor, so if this is the case you must say it. Therefore, it may not be necessarily an insecticide but a “mating disruptor” and not necessarily a killer of mosquitoes, but more like the ones that have been applied to disrupt mating in agricultural pests with pheromones. So, lines 388-391 are confusing or contradictory because if they are insecticides we do not care if disrupt reproduction because they will kill. You talk about resistance; one way to avoid resistance is to have two different modes of action at distinct target sites, so the probability of target site insensitivity is decreased. It is also a question of dose. Or just a cocktail of an insecticide with a mating disruptor? These hearing disruptors could be applied at lower dose with an insecticide of different mode of action? Re-write the paragraph 388-391 for clarity. The problem is line 388. And maybe instead of targets for insecticide development you must say “Novel targets for novel interventions is of highest priority...”

Thank you to the reviewer for these comments. We have modified the last paragraph in the manuscript to incorporate them (lines 448-464): *“We lastly tested whether the octopaminergic signalling in the mosquito ear could be targeted using pharmacological interventions to potentially interrupt mosquito partner recognition by disrupting mosquito hearing and thus eventually help control mosquito populations. We used the insecticide amitraz, an octopamine receptor agonist, as proof-of-concept of this approach. Amitraz is widely used as pesticide to kill ticks and parasitic mites 37,38, however it is not used as insecticide against mosquitoes probably due to its low toxicity 69. We found that amitraz exposure activates AgOct β 2 in the mosquito ear and induces the erection of the antennal fibrillae. It would be interesting to explore if amitraz, or preferably another compound acting as octopamine receptor antagonist -to mimic the auditory effects observed in AgOct β 2 knockout mosquitoes-, would disrupt mosquito mating by impairing the capacity of male mosquitoes to detect, or tune into, female flight tones. Such mating disruptors are promising novel tools for mosquito control 70,71 that could be delivered in the field using sugar baits 72. This would be of highest priority as the emergence of insecticide resistance and behavioural adaptations across malaria mosquitoes is endangering malaria control 73. Mating disruptors have been largely used for the control of insect pests 74. To implement them for mosquito control, more research is undoubtedly required to disentangle the neurological pathways involved in mosquito mating, identify suitable molecules to disrupt them and develop better delivery methods to target male mosquitoes. The exquisitely complex mosquito auditory system provides numerous opportunities to explore the potential of applying mating disruptors for mosquito control. ”*

Methods: can you please add the LIGHTS/lamps used in the Percival cabinets, what type of tubes were used? Intensity, anything that informs the light regime.

This information has been added (line 474-475): *“cycle (light intensity during the light cycle was 80 μ molm⁻² s⁻¹ from 4 fluorescent lamps)”*

L410 Eppendorfs with capital E.

This has been corrected.

It is not clear who prepared the Library (line 415). Was it done in house by authors or sent to some facility? Indicate.

This information has been added (line 429): *“The cDNA libraries were prepared by UCL genomics.”*

I cannot comment on the transcriptomic analyses or the formulas in pgs 13-14.

Sigma sells different p-octopamine so please add the catalog number and indicate if it was the analytical standard or what. Also, for clarity write para-octopamine hydrochloride (Sigma-Aldrich catalog #).

This has been added (line 632).

Similarly, for Amitraz I suggest writing the correct name or at least say if it was the Analytical standard and the NSC number below.

N'-(2,4-Dimethylphenyl)-N-[(2,4-dimethylphenyl)imino]methyl}-N-methylmethanimidamide, NSC 324552

This has been added (line 708).

Compound injection: was this done manually or with an injector apparatus?

I do not think the injection procedure is reproducible as described. If needed add a photograph as suppl information. I could not reproduce your description. What volume you injected approx.?

The injection was done using an injection setup controlled by a micromanipulator (see picture in Supplementary Figure 2) and a sharpened micro-capillary. We estimate that we injected aprox. 1-2µl of compound.

References

Check for consistency some have many authors some just say et al E.g number 52.

This has been corrected.

Ref 54 Belton has misspellings in two words.

This has been corrected.

Not sure about journal rules, but all scientific names of insects are not in italics and normally they should be.

We have italicized all scientific names in the references.

REVIEWERS' COMMENTS

Reviewer #1 (Remarks to the Author):

The authors have done an excellent job addressing all of my concerns. The conclusions are supported by the data, the methodology sound and the findings sufficiently novel. One of my requests was to know the identity of the two cycling genes in female antennae and the authors revealed that one of them was OrCo (the co-receptor for all tuning ORs). This is a potentially very important discovery and I strongly encourage the authors to follow-up on this observation in future experiments (but not this study, you're all good for this one).

Reviewer #2 (Remarks to the Author):

Yes, I read the response to reviewers, and all the figures are much more precise in color changes and the legends.

Most importantly my main concern was the lack of statistical analyses and now they included that in the figures and rearranged the comparisons so it makes much more sense when one reads the paper. They also took most of my suggestions and improved the text.

I believe the authors have addressed this reviewer's concerns and the manuscript is ready for publication.